# EMERGENCE IN NON-NEURAL MODELS: GROKKING MODULAR ARITHMETIC VIA AVERAGE GRADIENT OUTER PRODUCT

## ABSTRACT

Neural networks trained to solve modular arithmetic tasks exhibit *grokking*, the phenomenon where the test accuracy improves only long after the model achieves $100\%$ training accuracy in the training process. It is often taken as an example of "emergence", where model ability manifests sharply through a phase transition. In this work, we show that the phenomenon of grokking is not specific to neural networks nor to gradient descent-based optimization. Specifically, we show that grokking occurs when learning modular arithmetic with Recursive Feature Machines (RFM), an iterative algorithm that uses the Average Gradient Outer Product (AGOP) to enable task-specific feature learning with kernel machines. We show that RFM and, furthermore, neural networks that solve modular arithmetic learn block-circulant features transformations which implement the previously proposed Fourier multiplication algorithm.

## 1 INTRODUCTION

In recent years the idea of "emergence" has become an important narrative in machine learning. While there is no broad agreement on the definition (Rogers & Luccioni, 2023), it is often argued that "skills" emerge during the training process once certain data size, compute, or model size thresholds are achieved (Wei et al., 2022; Arora & Goyal, 2023). Furthermore, these skills are believed to appear rapidly, exhibiting sharp and seemingly unpredictable improvements in performance at these thresholds. One of the simplest and most striking examples supporting this idea is "grokking" modular arithmetic (Power et al., 2022; Nanda et al., 2023). A neural network trained to predict modular addition or another arithmetic operation on a fixed data set rapidly transitions from near-zero to perfect ($100\%$) test accuracy at a certain point in the optimization process. Surprisingly, this transition point occurs long after perfect *training accuracy* is achieved. Not only is this contradictory to the traditional wisdom regarding overfitting but, as we will show, some aspects of grokking do not fit neatly with our modern understanding of "benign overfitting" Bartlett et al. (2021); Belkin (2021).

Despite a large amount of recent work on emergence and, specifically, grokking, (see, e.g., (Power et al., 2022; Liu et al., 2023; Nanda et al., 2023; Thilak et al., 2022; Furuta et al., 2024; Miller et al., 2024)), the nature or even existence of the emergent phenomena remains contested. For example, the recent paper Schaeffer et al. (2023) suggests that the rapid emergence of skills may be a "mirage" due to the mismatch between the discontinuous metrics used for evaluation, such as accuracy, and the continuous loss used in training. The authors argue that, in contrast to accuracy, the test (or validation) loss or some other suitably chosen metric may decrease gradually throughout training and thus provide a useful measure of progress. Another possible progress measure is the training loss. As SGD-type optimization algorithms generally result in a gradual decrease of the training loss, one may posit that skills appear once the training loss falls below a certain threshold in the optimization process. Indeed, such a conjecture is in the spirit of classical generalization theory, which considers the training loss to be a useful proxy for the test performance Mohri et al. (2018).

In this work, we show that sharp emergence in modular arithmetic arises entirely from feature learning, independently of other aspects of modeling and training, and is not predicted by the standard measures of progress. We then clarify the nature of feature learning leading to the emergence of skills in modular arithmetic. We discuss these contributions in further detail below.

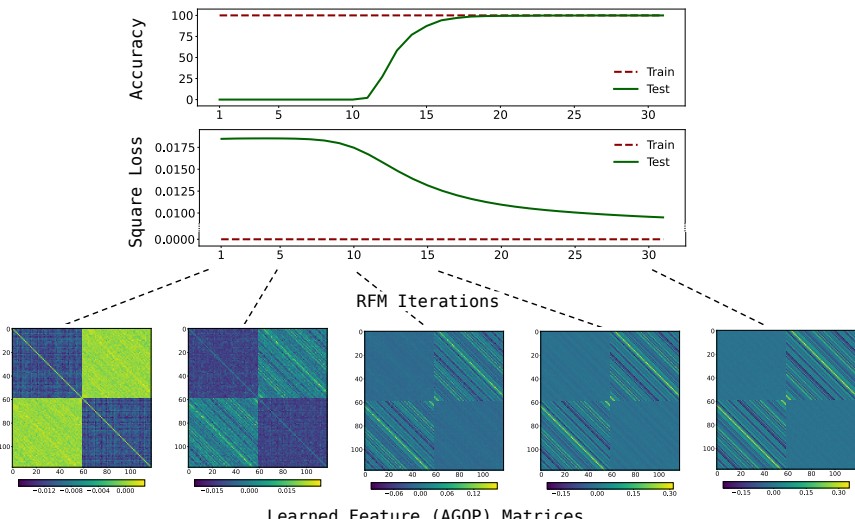

Figure 1: Recursive Feature Machines grok the modular arithmetic task $f^*(x, y) = (x + y) \bmod 59$.

**Summary of the contributions.** We demonstrate empirically that grokking modular arithmetic: (1) is not specific to neural networks; (2) is not tied to gradient-based optimization methods; (3) is not predicted by training or test loss[1], let alone accuracy.

Specifically, we show grokking for Recursive Feature Machines (RFM) (Radhakrishnan et al., 2024a), an algorithm that iteratively uses the Average Gradient Outer Product (AGOP) to enable task-specific feature learning in general machine learning models. In this work, we use RFM to enable feature learning in kernel machines, which are a class of predictors with no native mechanism for feature learning. In this setting, RFM iterates between three steps: (i) training a kernel machine, $f$, to fit training data; (ii) computing the AGOP matrix of $f$, $M$, over the training data to extract task-relevant features; and (iii) transforming input data, $x$, using the learned features via the map $x \rightarrow M^{s/2}x$ for a matrix power $s > 0$ (see Section 2 for details).

In Fig. 1 we give a representative example of RFM grokking modular addition, despite not using any gradient-based optimization methods and achieving perfect (numerically zero) training loss at every iteration. We see that during the first few iterations both the test loss and and test accuracy remain at the constant (random) level. Around iteration 10 the test loss starts improving and, a few iterations later, test accuracy quickly transitions to $100\%$. We also observe that even early in the iteration, structure emerges in AGOP feature matrices (see Fig. 1). The gradual appearance of structure in these feature matrices is striking given that the training loss is identically zero at every iteration and that the test loss does not significantly change until iteration 8. The striped patterns observed in feature matrices correspond to matrices whose sub-blocks are circulant with entries that are constant along the "long" diagonals which wrap around the matrix.[2] Such *circulant feature matrices* are key to learning modular arithmetic. In Section 3 we demonstrate that standard kernel machines using *random* circulant features easily learn modular operations. As these random circulant matrices are generic, we argue that no additional structure is required to solve modular arithmetic.

To demonstrate that the feature matrices evolve toward this structure (including for multiplication and division under an appropriate re-ordering of the input coordinates), we introduce two "hidden progress measures" (Barak et al., 2022): (1) *Circulant deviation*, which measures constancy of the diagonals of a matrix, and (2) *AGOP alignment*, which measures similarity between the feature matrix at iteration $t$ and the AGOP of a fully trained model. We will see that both of these measures show gradual (initially nearly linear) progress toward a model that generalizes.

---

[1]We note that for neural networks trained by SGD, emergence cannot be decoupled from training loss, as non-zero loss is required for training to occur at all.

[2]Feature sub-matrices may also be constant on anti-diagonals. We also refer to these matrices as circulant.

We further argue that emergence in fully connected neural networks trained on modular arithmetic identified in prior work (Gromov, 2023; Liu et al., 2022) is analogous to that for RFM and is exhibited through the AGOP (see Section 4). By visualizing covariances of network weights, we observe that these models also learn block-circulant features to grok modular arithmetic. We demonstrate that these features are highly correlated with the AGOP of neural networks, corroborating prior observations from Radhakrishnan et al. (2024a). Furthermore, paralleling our observations for RFM, our progress measures indicate gradual progress toward a generalizing solution during neural network training. Finally we demonstrate that training neural networks on data transformed by random block-circulant matrices dramatically decreases training time needed to learn modular arithmetic.

Why are these learned block circulant features effective for modular arithmetic? We provide supporting theoretical evidence that circulant features result in kernel machines implementing the Fourier Multiplication Algorithm (FMA) for modular arithmetic (see Section 5). For the case of neural networks, several prior works have argued empirically and theoretically that neural networks learn to implement the FMA to solve modular arithmetic (Nanda et al., 2023; Varma et al., 2023; Morwani et al., 2024). While kernel RFM and neural networks utilize different classes of predictive models, our results suggest that they discover similar algorithms for implementing modular arithmetic.

By decoupling feature learning from predictor training, our results provide evidence for emergent properties of machine learning models arising purely as a consequence of their ability to learn features. We hope our work will help isolate the underlying mechanisms of emergence and shed light on the key practical concern of how, when, and why these seemingly unpredictable transitions occur.

**Paper outline.** Section 2 reviews preliminary concepts. In Section 3, we demonstrate emergence with RFM and show AGOP features consist of circulant blocks. Section 4, shows that neural network features are circulant and are captured by the AGOP. In Section 5, we prove that kernel machines learn the FMA with circulant features. We provide a discussion and conclude in Section 6.

## 2 PRELIMINARIES

**Learning modular arithmetic.** Let $\mathbb{Z}_p = \mathbb{Z}/p\mathbb{Z}$ denote the field of integers modulo a prime $p$ and let $\mathbb{Z}_p^* = \mathbb{Z}_p \setminus \{0\}$. We learn modular functions $f^*(a, b) = g(a, b) \bmod p$ where $f^* : \mathbb{Z}_p \times \mathbb{Z}_p \to \mathbb{Z}_p$, $a, b \in \mathbb{Z}_p$, and $g : \mathbb{Z} \times \mathbb{Z} \to \mathbb{Z}$ is an arithmetic operation on $a$ and $b$, e.g. $g(a, b) = a + b$. Note that there are $p^2$ discrete input pairs $(a, b)$ for all modular operations except for $f^*(a, b) = (a \div b) \bmod p$, which has $p(p-1)$ inputs as the denominator cannot be 0.

To train models on modular arithmetic tasks, we construct input-label pairs by one-hot encoding the input and label integers. Specifically, for every pair $a, b \in \mathbb{Z}_p$, we write the input as $\boldsymbol{e}_a \oplus \boldsymbol{e}_b \in \mathbb{R}^{2p}$ and the output as $\boldsymbol{e}_{f^*(a,b)} \in \mathbb{R}^p$, where $\boldsymbol{e}_i \in \mathbb{R}^p$ is the $i$-th standard basis vector in $p$ dimensions and $\oplus$ is concatenation. The training dataset consists of a random subset of $n = r \times N$ input/label pairs, where $r$ is the *training fraction* and $N = p^2$ or $p(p-1)$ is the number of possible discrete inputs.

**Circulant matrices.** The features that RFMs and neural networks learn in order to solve modular arithmetic contain blocks of *circulant matrices*, which are defined as follows. Let $\sigma : \mathbb{R}^p \to \mathbb{R}^p$ be the cyclic permutation which acts on a vector $u \in \mathbb{R}^p$ by shifting its coordinates by one cell to the right: $[\sigma(u)]_j = u_{j-1 \bmod p}$, for $j \in [p]$. We write the $\ell$-fold composition of this map $\sigma^\ell(u) \in \mathbb{R}^p$ with entries $[\sigma^\ell(u)]_j = u_{j-\ell \bmod p}$. A circulant matrix $C \in \mathbb{R}^{p \times p}$ is determined by a vector $\boldsymbol{c} = [c_0, \ldots, c_{p-1}] \in \mathbb{R}^p$, and has rows (in order from first to last): $\boldsymbol{c}, \sigma(\boldsymbol{c}), \ldots, \sigma^{p-1}(\boldsymbol{c})$. Feature matrices may also have have constant anti-diagonals (so-called Hankel matrices). To ease terminology, we will use the word circulant to refer to both Hankel and circulant matrices.

**Average Gradient Outer Product (AGOP).** The AGOP matrix, which will be central to our discussion, is defined as follows.

**Definition 2.1** (AGOP). *Given a predictor* $f : \mathbb{R}^d \to \mathbb{R}^c$ *with* $c$ *outputs,* $f(x) \equiv [f_0(x), \ldots, f_{c-1}(x)]$, *let* $\frac{\partial f(x')}{\partial x} \in \mathbb{R}^{d \times c}$ *be the Jacobian (transposed) of* $f$ *evaluated at some point* $x' \in \mathbb{R}^d$ *with entries* $[\frac{\partial f(x')}{\partial x}]_{s,\ell} = \frac{\partial f_\ell(x')}{\partial x_s}$. *Then, for* $f$ *trained on a set of data points* $\{x^{(j)}\}_{j=1}^n$, *with* $x^{(j)} \in \mathbb{R}^d$, *the Average Gradient Outer Product (AGOP), G, is defined as,*

$$G(f; \{x^{(j)}\}_{j=1}^n) = \frac{1}{n} \sum_{j=1}^n \frac{\partial f(x^{(j)})}{\partial x} \frac{\partial f(x^{(j)})}{\partial x}^\top \in \mathbb{R}^{d \times d}. \qquad (1)$$

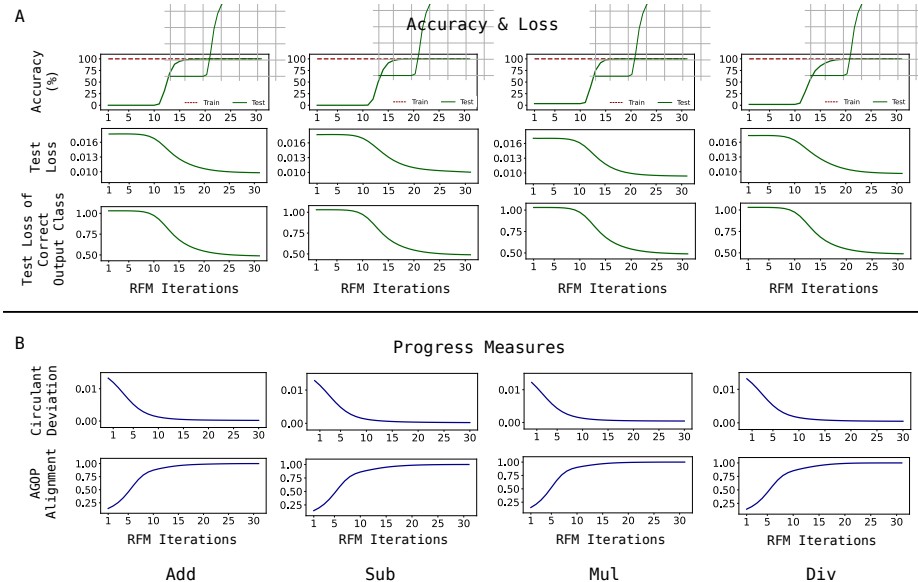

Figure 2: RFM with the quadratic kernel on modular arithmetic with modulus $p = 61$ trained for 30 iterations. (A) Test accuracy, test loss (mean squared error) over all output coordinates, and test loss of the correct class output coordinate do not change in the first 8 iterations and then, sharply transition. (B) Circulant deviation and AGOP alignment show gradual progress towards generalizing solutions despite accuracy and loss metrics not changing in the initial iterations. For multiplication (Mul) and division (Div), circulant deviation is measured with respect to the feature sub-matrices after reordering by the discrete logarithm.

For simplicity, we omit the dependence on the dataset in the notation. Top eigenvectors of AGOP can be viewed as the "most relevant" input features, those input directions that influence the output of a general predictor (for example, a kernel machines or a neural network) the most. As a consequence, the AGOP can be viewed as a task-specific transformation that can be used to amplify relevant features and improve sample efficiency of machine learning models.

Indeed, a line of prior works (Yuan et al., 2023; Trivedi et al., 2014; Hristache et al., 2001) have used the AGOP to improve the sample efficiency of predictors trained on multi-index models, a class of predictive tasks in which the target function depends on a low-rank subspace of the data. Though the study of AGOP has been motivated by these multi-index examples, we will see that the AGOP can be used to recover useful features for modular arithmetic that are, in fact, not low-rank.

**AGOP and feature learning in neural networks.** Radhakrishnan et al. (2024a) posited that AGOP was a mechanism through which neural networks learn features. In particular, the authors introduce the *Neural Feature Ansatz (NFA)* stating that for any layer $\ell$ of a trained neural network with weights $W_\ell$, the *Neural Feature Matrix (NFM)*, $W_\ell^T W_\ell$, are highly correlated to the AGOP of the model computed with respect to the input of layer $\ell$. The NFA suggests that neural networks learn features at each layer by utilizing the AGOP. For more details on the NFA, see Appendix C.

**Recursive Feature Machine (RFM).** Importantly, AGOP can be computed for any differentiable predictor, including those such as kernel machines that have no native feature learning mechanism. As such, the authors of Radhakrishnan et al. (2024a) developed an algorithm known as RFM, which iteratively uses the AGOP to extract features. Below, we present the RFM algorithm used in conjunction with kernel machines. Suppose we are given data samples $(X, y) \in \mathbb{R}^{n \times d} \times \mathbb{R}^n$ where $X$ contains $n$ samples denoted $\{x^{(j)}\}_{j=1}^n$. Given an initial symmetric positive-definite matrix $M_0 \in \mathbb{R}^{d \times d}$, and Mahalanobis kernel $k(\cdot, \cdot; M) : \mathbb{R}^d \times \mathbb{R}^d \to \mathbb{R}$, RFM iterates the following steps for $t \in [T]$:

*Step 1 (Predictor training):* $f^{(t)}(x) = k(x, X; M_t)\alpha$ with $\alpha = k(X, X; M_t)^{-1}y$ ; $\qquad$ (2)

*Step 2 (AGOP update):* $M_{t+1} = [G(f^{(t)})]^s$ ; $\qquad$ (3)

where $s > 0$ is a matrix power and $k(X, X; M) \in \mathbb{R}^{n \times n}$ denotes the matrix with entries $[k(X, X; M)]_{j_1 j_2} = k(x^{(j_1)}, x^{(j_2)}; M)$ for $j_1, j_2 \in [n]$. In this work, we select $s = \frac{1}{2}$ for all experiments (see Algorithm 1 for complete pseudocode). We use the following two Mahalanobis kernels: (1) the quadratic kernel, $k(x, x'; M) = \left(x^\top M x'\right)^2$; and (2) the Gaussian kernel $k(x, x'; M) = \exp\left(-\|x - x'\|_M^2 / L\right)$, where for $z \in \mathbb{R}^d$, $\|z\|_M^2 = z^\top M z$, and $L$ is the bandwidth.

# 3    EMERGENCE WITH RECURSIVE FEATURE MACHINES

We now show that RFM exhibits sharp transitions in performance on modular arithmetic tasks (addition, subtraction, multiplication, and division) due to the emergence of block-circulant features.

We will use a modulus of $p = 61$ and train RFM with quadratic and Gaussian kernel machines (experimental details are provided in Appendix D). As we solve kernel ridgeless regression exactly, all iterations of RFM result in zero training loss and 100% training accuracy. The top two rows of Fig. 2A show that the first several iterations of RFM result in near-zero test accuracy and approximately constant, large test loss. Despite these standard progress measures initially not changing, continuing to iterate RFM leads to a dramatic, sharp increase to 100% test accuracy and a corresponding decrease in the test loss later in the iteration process.

**Sharp transition in loss of correct output coordinate.** It is important to note that our total loss function is the square loss averaged over $p = 61$ classes. It is thus plausible that, due to averaging, the near-constancy of the total square loss over the first few iterations conceals steady improvements in the predictions of the correct class. However, in Fig. 2A (third row) we show that the test loss for the output coordinate (logit) of the correct class closely tracks the total test loss.

**Emergence of block-circulant features in RFM.** To understand RFM generalization, we visualize the $2p \times 2p$ feature matrix given by the square root of the AGOP from the final iteration of RFM. We first visualize the feature matrices for RFM trained on modular addition/-subtraction in Fig. 3A. Their visually-evident striped structure suggests a more precise characterization:

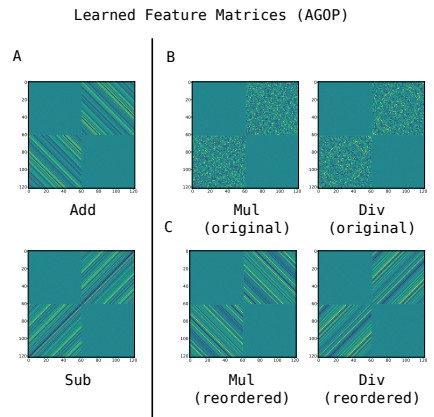

Learned Feature Matrices (AGOP)

Figure 3: RFM with the quadratic kernel for modular arithmetic with $p = 61$. (A) The square root of the kernel AGOPs for addition (Add), subtraction (Sub) visualized without their diagonals to emphasize the off-diagonal blocks. (B) Square root of the kernel AGOP for multiplication (Mul), division (Div). (C) For Mul and Div, rows and columns of each sub-matrix is re-ordered by the discrete log. base 2.

**Observation 1** (Block-circulant features). *Feature matrix $M^* \in \mathbb{R}^{2p \times 2p}$ at the final iteration of RFM on modular addition/subtraction is of the form*

$$M^* = \begin{pmatrix} A & C^\top \\ C & A \end{pmatrix},\tag{4}$$

*where $A, C \in \mathbb{R}^{p \times p}$, $C$ is an asymmetric circulant matrix. , $A = c_1 I + c_2 \mathbf{1}\mathbf{1}^\top$ for scalars $c_1, c_2$.*

Similarly to addition and subtraction, RFM successfully learns multiplication and division. Yet, in contrast to addition and subtraction, the structure of feature matrices for these tasks, shown in Fig. 3B, is not at all obvious. Nevertheless, re-ordering the rows and columns of the feature matrices for these tasks brings out their hidden circulant structure of the form stated in Eq. (4). We show the effect of re-ordering in Fig. 3C (see also Appendix Fig. 1 for the evolution of re-ordered and original features during training).

We briefly discuss the reordering procedure below and provide further details in Appendix E. To reorder, we use the fact of group theory that the multiplicative group $\mathbb{Z}_p^*$ is a cyclic group of order $p - 1$ (e.g., Koblitz (1994)). By definition of the cyclic group, there exists at least one element $g \in \mathbb{Z}_p^*$, known as a *generator*, such that $\mathbb{Z}_p^* = \{g^i \; ; \; i \in \{1, \ldots, p - 1\}\}$. As we will see, re-ordering the rows and columns of the AGOP by powers of a generator reveals circulant structure.

For modular multiplication/division, the map taking $g^i$ to $i$ is known as the *discrete logarithm* base $g$ (Koblitz, 1994, Ch.3). It is natural to expect block-circulant feature matrices to arise in modular multiplication/division after reordering by the discrete log as the discrete log converts modular multiplication/division into modular addition/subtraction. We note the recent work Doshi et al. (2024) also used the discrete log to reorder coordinates in the context of constructing a solution for solving modular multiplication with neural networks.

**Progress measures.** We propose and examine two measures of feature learning, *circulant deviation* and *AGOP alignment*.

*Circulant deviation.* As the final feature matrices contain circulant sub-blocks, a natural progress measure for learning modular arithmetic with RFM is how far AGOP feature matrices are from a block-circulant matrix. For a feature matrix $M$, let $A$ denote the bottom-left sub-block of $M$. We define circulant deviation as the total variance of the (wrapped) diagonals of $A$ normalized by the norm $\|A\|_F^2$. In particular, let $\mathcal{S} \in \mathbb{R}^{p \times p} \to \mathbb{R}^{p \times p}$ denote the shift operator, which shifts the $\ell$-th row of the matrix by $\ell$ positions to the right. Also let $\mathrm{Var}(\boldsymbol{v}) = \sum_{j=0}^{p-1}(v_j - \mathbb{E}\boldsymbol{v})^2$ be the variance of a vector $\boldsymbol{v}$. If $A[j]$ denotes the $j$-th column of $A$, we define circulant deviation $\mathcal{D}$ as: $\mathcal{D}(A) = \frac{1}{\|A\|_F^2} \sum_{j=0}^{p-1} \mathrm{Var}(\mathcal{S}(A)[j])$. As circulant matrices are constant along their (wrapped) diagonals, they have a circulant deviation of $0$.

We see in Fig. 2B (top row) that circulant deviation exhibits gradual improvement through the course of training with RFM. We find that for the first 10 iterations, while the training loss is numerically zero and the test loss does not improve, circulant deviation exhibits gradual, nearly linear, improvement. The improvements in circulant deviation reflect visual improvements in features, as was also shown in Fig. 1. These curves also provide further support for Observation 1, as the circulant deviation is close to $0$ at the end of training.

Circulant deviation depends crucially on the observation that for modular arithmetic the feature matrices contained circulant blocks. For more general tasks, we may not be able to identify such structure. Thus, we propose a second, more general progress measure, AGOP alignment.

*AGOP alignment.* Given two matrices $A, B \in \mathbb{R}^{d \times d}$, let $\rho(A, B)$ denote the standard cosine similarity between these two matrices when vectorized. Specifically, let $\tilde{A}, \tilde{B} \in \mathbb{R}^{d^2}$ denote the vectorization of $A$ and $B$ respectively, then $\rho(A, B) = \frac{\langle \tilde{A}, \tilde{B} \rangle}{\|\tilde{A}\| \|\tilde{B}\|}$.

If $M_t$ denotes the AGOP at iteration $t$ of RFM (or epoch $t$ of a neural network) and $M^*$ denotes the final AGOP of the trained RFM (or neural network), then AGOP alignment at iteration $t$ is given by $\rho(M_t, M^*)$. The same measure of alignment was used in Zhu et al. (2024), except their alignment was computed with respect to the AGOP of the ground truth model. Note that as modular operations are discrete, in our setting there is no unique ground truth model for which AGOP can be computed.

Like circulant deviation, AGOP alignment exhibits gradual improvement in the regime that test loss is constant and large (see Fig. 2B, bottom row). Moreover, AGOP alignment is a more general progress measure since it does not require assumptions on the structure of the AGOP. For instance, AGOP alignment can be measured without reordering for modular multiplication/division. While AGOP alignment does not require a specific form of the final features, it is still an *a posteriori* measurement of progress as it requires access to the features of a fully trained model.

**Random circulant features allow standard kernels to generalize.** We conclude this section by providing further evidence that the form of feature matrices given in Observation 1 is key to enabling generalization in kernel

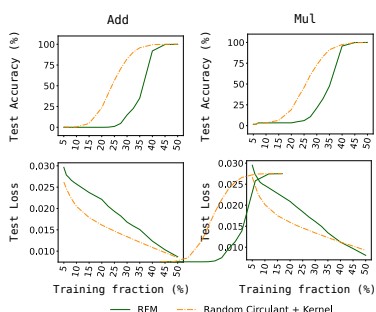

Figure 4: Random circulant features generalize with standard kernels for modular arithmetic. RFM with the Gaussian kernel on addition (Add) and multiplication (Mul) for modulus $p = 61$ is compared to a base Gaussian kernel machine trained on random circulant features (for Mul, the sub-blocks are circulant after re-ordering by the discrete logarithm base 2).

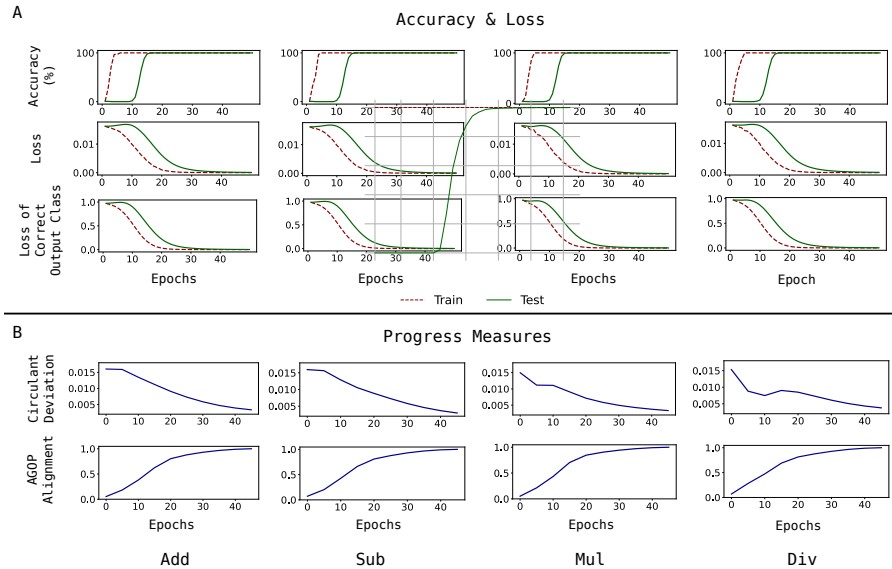

Figure 5: One hidden layer fully-connected networks with quadratic activations trained on modular arithmetic with $p = 61$ trained for 50 epochs with the square loss. (A) Test accuracy, test loss over all outputs, and test loss of the correct class output do not change in the initial iterations. (B) Progress measures for circulant deviation and AGOP alignment. Circulant deviation for Mul and Div are computed after reordering by the discrete logarithm base 2.

machines trained to solve modular arithmetic tasks. We now show that a transformation with a *generic* block-circulant matrix enables kernels machines to learn modular arithmetic. We generate a random circulant matrix $C$ by first sampling entries of the first column i.i.d. from the uniform distribution on $[0, 1] \subset \mathbb{R}$ and then shifting the column to generate the remaining columns of $C$. We construct $M^*$ in Observation 1 with $c_1 = 1, c_2 = -1/p$. For modular addition, we transform the input data by mapping $x_{ab} = e_a \oplus e_b$ to $\tilde{x}_{ab} = (M^*)^{\frac{1}{4}} x_{ab}$, and then train on the new data pairs $(\tilde{x}_{ab}, e_{a+b \bmod p})$ for a subset of all possible pairs $(a, b) \in \mathbb{Z}_p^2$. Note that transforming data with $(M^*)^{\frac{1}{4}}$ is akin to using $s = 1/2$ in RFM.

We do the same for modular multiplication after reordering the random circulant by the discrete logarithm as described above. The experiments in Fig. 4 show that standard kernel machines trained on feature matrices with random circulant blocks outperform RFM that learns such features through AGOP. We also find that directly enforcing circulant blocks in the sub-matrices of $M_t$ throughout RFM iterations accelerates grokking and improves test loss (see Appendix F, Appendix Fig. 2). These experiments provide direct evidence that the structure in Observation 1 is key for generalization on modular arithmetic and, furthermore, *no additional structure* beyond a generic circulant is required.

## 4 EMERGENCE IN NEURAL NETWORKS THROUGH AGOP

We now show that grokking in two-layer neural networks relies on the same principles as grokking by RFM. Specifically we demonstrate that (1) block-circulant features are key to neural networks grokking modular arithmetic; and (2) our measures (circulant deviation and AGOP alignment) indicate gradual progress towards generalization, while standard measures of generalization exhibit sharp transitions. All experimental details are provided in Appendix D.

**Grokking with neural networks.** We first reproduce grokking with modular arithmetic using fully-connected networks as identified in prior works (Fig. 5A) (Gromov, 2023). In particular, we train one hidden layer fully connected networks $f : \mathbb{R}^{2p} \to \mathbb{R}^p$ of the form $f(x) = W_2 \phi(W_1 x)$ with quadratic activation $\phi(z) = z^2$ on modulus $p = 61$ data with a training fraction 50%.

Consistent with prior work (Gromov, 2023) and analogously to RFMs, neural networks exhibit an initial training period where the train accuracy reaches 100%, while test accuracy is at 0% and test

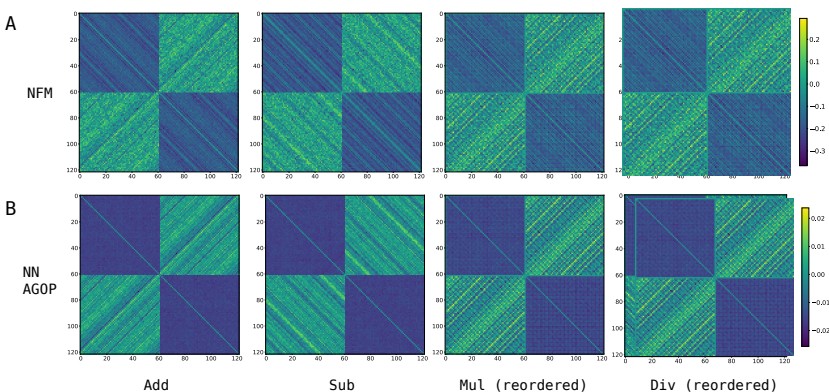

Figure 6: Feature matrices from one hidden layer neural networks with quadratic activations trained on addition, subtraction, multiplication, and division modulo 61. The Pearson correlations between the NFM and square root of the AGOP for each task are 0.955 (Add), 0.942 (Sub), 0.924 (Mul), 0.929 (Div). Mul and Div are shown after reordering by the discrete logarithm base 2.

loss does not improve (see Fig. 5A). After this point, we see that the accuracy rapidly improves to achieve perfect generalization. We further verify that the sharp transition in test loss is not an artifact of averaging the loss over all output coordinates. In the third row of Fig. 5A we show that the test loss of the individual correct output coordinate closely tracks the total loss.

**Emergence of block-circulant features in neural networks.** To understand the features learned by neural networks we visualize the first layer Neural Feature Matrix, defined as follows.

**Definition 4.1.** *Given a fully connected network $f(x) = a^\top \phi(W_1 x)$, the first layer Neural Feature Matrix (NFM) is the matrix $W_1^\top W_1 \in \mathbb{R}^{2p \times 2p}$.*

The NFM is the un-centered covariance of network weights and has been used in prior work in order to understand the features learned by various neural network architectures at any layer (Radhakrishnan et al., 2024a; Trockman et al., 2022). Fig. 6A displays the NFM for one hidden layer neural networks with quadratic activations trained on modular arithmetic tasks. For addition/subtraction, we find that the NFM exhibits block circulant structure, akin to the feature matrix for RFM. As described in Section 3 and Appendix E, we reorder the NFM for networks trained on multiplication/division with respect to a generator for $\mathbb{Z}_p^*$ in order to observe block-circulant structure (see Appendix Fig. 4A for a comparison of multiplication/division NFMs before and after reordering). The block-circulant structure in both the NFM and the feature matrix of RFM suggests that the two models are learning similar sets of features.

The work Radhakrishnan et al. (2024a) posited that AGOP is the mechanism through which neural networks learn features. The authors stated their claim in the form of the Neural Feature Ansatz (NFA), which states that NFMs are proportional to a matrix power of AGOP through training (see Eq. (5) for a restatement of the NFA). As such, we additionally compute the square root of the AGOP to examine the features learned by neural networks trained on modular arithmetic tasks. We visualize the square root of the AGOPs of these trained models in Fig. 6B and also find that the square root of the AGOP and the NFM are highly correlated (greater than 0.92), where Pearson correlation is equal to cosine similarity after centering the inputs to be mean 0. Moreover, we find that the square root of AGOP of neural networks again exhibits the same structure as stated in Observation 1 (see Appendix Fig. 4B for a comparison of multiplication/division AGOPs before and after reordering).

**Random circulant maps improve generalization of neural networks.** To further establish the importance and generality of block-circulant features, we demonstrate that training networks on inputs transformed with a random block-circulant matrix greatly accelerates learning. In Fig. 7, we compare the performance of neural networks trained on one-hot encoded modulo $p$ integers and the same integers transformed with a random block-circulant matrix. At a training fraction of 17.5%, we find that networks trained on transformed integers achieved 100% test accuracy within several hundred epochs and exhibit little delayed generalization while networks trained on non-transformed integers do not achieve 100% test accuracy even within 3000 epochs.

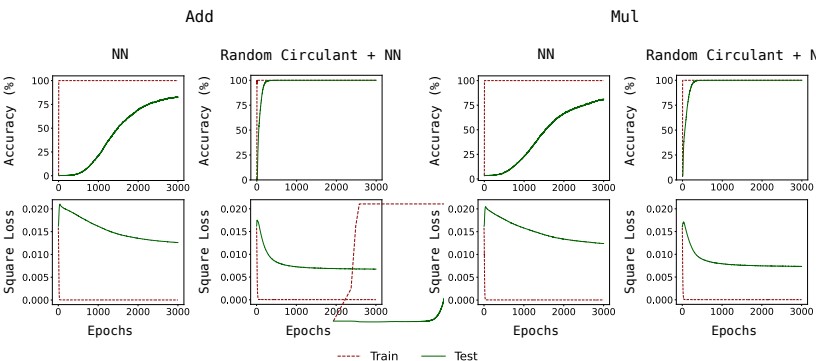

Figure 7: Random circulant features speed up generalization in neural networks for modular arithmetic tasks. We compare one hidden layer MLPs with quadratic activations trained on modular addition and multiplication for $p = 61$ using standard one-hot encodings or those transformed by random circulant matrices (re-ordered by the discrete logarithm for multiplication).

**Progress measures.** Given that the square root of the AGOP of neural networks exhibits block-circulant structure, we can use circulant deviation and AGOP alignment to measure gradual progress of neural networks toward a generalizing solution. As before, we measure circulant deviation in the case of multiplication/division after reordering the feature submatrix by a generator of $\mathbb{Z}_p^*$. In Fig. 5B, we see that our measures indicate gradual progress in contrast to sharp transitions in the standard measures of progress shown in Fig. 5A. There is a period of 5-10 epochs where circulant deviation and AGOP alignment improve while test loss and test accuracy do not. As was the case of RFM, these metrics reveal gradual progress of neural networks toward generalizing solutions.

## 5 FOURIER MULTIPLICATION ALGORITHM FROM CIRCULANT FEATURES

We have seen so far that features containing circulant sub-blocks enable generalization for RFMs and neural networks across modular arithmetic tasks. We now provide theoretical support that shows how kernel machines equipped with such circulant features learn generalizing solutions. In particular, we show that there exist block-circulant feature matrices, as in Observation 1, such that kernel machines equipped with these features and trained on all available data for a given modulus $p$ solve modular arithmetic through the *Fourier Multiplication Algorithm* (FMA). Notably, the FMA has been argued both empirically and theoretically in prior works to be the solution found by neural networks to solve modular arithmetic (Nanda et al., 2023; Zhong et al., 2024).

The FMA is a specific solution for implementing modular arithmetic that first represents the data by its Discrete Fourier Transform (DFT). Intuitively, transforming the data with circulant matrices extracts the DFT of the one-hot encoded vectors following the well-known fact that circulant matrices can be diagonalized using the matrix that encodes the DFT (Gray et al., 2006). We state our result informally here (for more details on the FMA, the precise theorem, and its proof, see Appendix G).

**Theorem 5.1** (Circulant features give the FMA). *Training on all of the discrete data for any modular operation, for each output class $\ell \in \{0, \cdots, p-1\}$, suppose we train a separate quadratic kernel predictor and particular block-circulant feature matrices $M_\ell$ (having the structure in Observation 1). Then, the concatenated predictor given by kernel ridgeless regression on each output is equivalent to the Fourier Multiplication Algorithm for that modular operation.*

Notably, the FMA is defined over all of $\mathbb{R}^{2p}$, not just on one-hot encoded inputs. Thus, not only do neural networks and RFM learn similar features, we have established a setting where kernel methods equipped with block-circulant feature matrices learn the same out-of-domain solution as neural networks for these tasks. This result is interesting, in part, as the only constraint for generalization on these tasks is to obtain perfect accuracy on inputs that are standard basis vectors.

## 6 DISCUSSION AND CONCLUSIONS

Most classical analyses of generalization relied on the training loss serving as a proxy for the test loss and thus a useful measure of generalization. Empirical results of deep learning have upended this

long-standing belief. In many settings, predictors that fit the data exactly can still generalize, thus invalidating training loss as a predictor of test performance. This has led to the recent developments in understanding benign overfitting, in neural networks as well as in classical kernel and linear models Belkin (2021); Bartlett et al. (2021). Since the training loss may not predict generalization, the common suggestion has been to use the validation loss computed on a separate *validation dataset*. Emergent phenomena, such as grokking, show that we cannot rely even on validation performance at intermediate training steps to predict generalization at the end of training. Indeed, validation loss at a certain iteration may not be indicative of the validation loss itself only a few iterations later. Further, contrary to Schaeffer et al. (2023), we show these phase transitions in performance are not generally "a mirage" since, as we observe in this work, they are not always predicted by *a priori* measures of performance, continuous or discontinuous. Instead, emergence is fully determined by feature learning, which is difficult to observe without having access to a fully trained model. Indeed, the progress measures discussed in this work, as well as those suggested in, e.g., Barak et al. (2022); Nanda et al. (2023); Doshi et al. (2024) can be termed *a posteriori* progress indicators. They all require either understanding of the algorithm implemented by a generalizing trained model (such as our circulant deviation, the Fourier gap considered in Barak et al. (2022), or the Inverse Participation Ratio in Doshi et al. (2024)) or access to such a model (e.g. AGOP alignment).

Consider generalizing features for modular multiplication shown in Fig. 3. The original features shown in panel B of this figure do not have an easily identifiable pattern. In contrast, re-ordered features in panel C are clearly striped, containing block-circulants. As discussed in Section 3, re-ordering of features requires understanding that the multiplicative group $\mathbb{Z}_p^*$ is cyclic of order $p - 1$. While a well-known result, it is far from obvious *a priori*. It is thus plausible that in other settings hidden feature structures may be hard to identify due to a lack of mathematical insight.

**Why is learning modular arithmetic surprising?** The task of learning modular operations is different from many other statistical machine learning tasks. In continuous ML settings, we typically posit that the "ground truth" target function is smooth in an appropriate sense. Hence any general purpose algorithm capable of learning smooth functions (such as, for example, $k$-nearest neighbors) should be able to learn the target function given enough data. Primary differences between learning algorithms are thus in sample and computational efficiency. In contrast, it is unclear what principle leads to learning modular arithmetic from partial observations. There are many ways to fill in the missing data and we do not know a simple inductive bias, to guide us toward a solution. Several recent works argued that margin maximization with respect to certain norms can account for learning modular arithmetic (Morwani et al., 2024; Lyu et al., 2023; Mohamadi et al., 2024). While the direction is promising, general underlying principles are not yet clear.

**Analyses of grokking.** Recent works (Kumar et al., 2024; Lyu et al., 2023; Mohamadi et al., 2024) argue that grokking occurs in neural networks through a two phase mechanism that transitions from a "lazy" regime, with no feature learning, to a "rich" feature learning regime. Our experiments clearly show that grokking in RFM does not undergo such a transition. For RFM on modular arithmetic tasks, our progress measures indicate that the features evolve gradually toward the final circulant matrices, even as test performance initially remains constant (Fig. 2). Grokking in these settings is entirely due to the gradual feature quality improvement and two-phase grokking does not occur. Additionally, we have not observed significant evidence of "lazy" to "rich" transition as a mechanism for grokking in our experiments with neural networks, as most of our measures of feature learning start improving early on in the training process (improvement in circulant deviation measure is delayed for addition and subtraction, but not for multiplication and division, while AGOP feature alignment initially shows near linear improvement for all tasks), see Fig. 5. Our observations for neural networks are in line with the results in (Doshi et al., 2024; Nanda et al., 2023), where their proposed progress measures, Inverse Participation Ratio and Gini coefficients of the weights in the Fourier domain, are shown to increase prior to improvements in test loss and accuracy for modular arithmetic.

**Conclusions.** In this paper, we showed that grokking modular arithmetic happens in feature learning kernel machines in a manner very similar to what has been observed in neural networks. Remarkably we observe that feature learning can happen independently of improvements in both training and test loss. Not only does this finding reinforce the narrative of rapid emergence of skills in neural networks, it is also not easily explicable within the framework of the existing generalization theory.

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

---

**Algorithm 1** Recursive Feature Machine (RFM) (Radhakrishnan et al., 2024a)

---

**Require:** $X, y, k, T, L$        $\triangleright$ Train data: $(X, y)$, base kernel: $k$, iters.: $T$, matrix power: $s$, and
    bandwidth: $L$
    $M_0 = I_d$
    **for** $t = 0, \ldots, T-1$ **do**
        Solve $\alpha \leftarrow k(X, X; M_t)^{-1} y$                $\triangleright f^{(t)}(x) = k(x, X; M_t)\alpha$
        $M_{t+1} \leftarrow [G(f^{(t)})]^s$
    **end for**
    **return** $\alpha, M_{T-1}$          $\triangleright$ Solution to kernel regression: $\alpha$, and feature matrix: $M_{T-1}$

---

## A   BROADER DISCUSSION

**Low rank learning.** The problem of learning modular arithmetic can be viewed as a type of matrix completion – completing the $p \times p$ matrix (so-called Cayley table) representing modular operations, from partial observations. The best studied matrix completion problem is low rank matrix completion, where the goal is to fill in missing entries of a low rank matrix from observing a subset of the entries (Moitra, 2018, Ch.8). While many specialized algorithms exist, it has been observed that neural networks can recover low rank matrix structures Gunasekar et al. (2017). Notably, in a development paralleling the results of this paper, low-rank matrix completion can provably be performed by linear RFMs using the same AGOP mechanism Radhakrishnan et al. (2024b).

It is thus tempting to posit that grokking modular operations in neural networks or RFM can be explained as a low rank prediction problem. Indeed modular operations can be implemented by an index 4 model, i.e., a function of the form $f = g(Ax)$, where $x \in \mathbb{R}^{2p}$ and $A$ is a rank 4 matrix (see Appendix L for the construction). It is a plausible conjecture as there is strong evidence, empirical and theoretical, that neural networks are capable of learning such multi-index models Damian et al. (2022); Mousavi-Hosseini et al. (2022) as well as low-rank matrix completion. Furthermore, a phenomenon similar to grokking was discussed in (Radhakrishnan et al., 2022, Fig. 5, 6) in the context of low rank feature learning for both neural networks and RFM. However, despite the existence of generalizeable low rank models, the actual circulant features learned by both Neural Networks and RFM are *not* low rank. Interestingly, this observation mirrors the problem of learning parity functions through neural network inspired minimum norm interpolation, which was analyzed in Ardeshir et al. (2023). While single-directional (index one) solutions exist in that setting, the authors show that the minimum norm solutions are all multi-dimensional.

**Explanations for deep learning** Finally, this work adds to the growing body of evidence that the AGOP-based mechanisms of feature learning can account for some of the most interesting phenomena in deep learning. These include generalization with multi-index models (Parkinson et al., 2023), deep neural collapse (Beaglehole et al., 2024b), and the ability to perform low-rank matrix completion (Radhakrishnan et al., 2024b). Thus, RFM provides a framework that is both practically powerful and serves as a theoretically tractable model of deep learning.

## B   ADDITIONAL PRELIMINARIES

For completeness we replicate the algorithm definition for Recursive Feature Machines (RFM) provided by Radhakrishnan et al. (2024a) in Algorithm 1. This procedure recursively fits a kernel estimator for a chosen base kernel, $k$, then updates the feature matrix, $M$, by computing a matrix power of the Average Gradient Outer Product (AGOP) for that estimator. The algorithm terminates after a total of $T$ iterations. The final estimator and feature matrix are then returned by the algorithm.

## C   NEURAL FEATURE ANSATZ

While the NFA has been observed generally across depths and architecture types (Radhakrishnan et al., 2024a; Beaglehole et al., 2023; 2024a), we restate this observation for fully-connected networks with one hidden-layer of the form $f(x) = a^\top \phi(W_1 x)$.

**Ansatz 1** (Neural Feature Ansatz for one hidden layer). *For a one hidden-layer neural network $f^{\mathrm{NN}}$ and a matrix power $\alpha \in (0, 1]$, the following holds:*

$$W_1^\top W_1 \propto G(f^{\mathrm{NN}})^s . \tag{5}$$

Note that this statement implies that $W_1^\top W_1$ and $G(f^{\mathrm{NN}})^s$ have a cosine similarity of $\pm 1$.

In this work, we choose $\alpha = \frac{1}{2}$, following the main results in Radhakrishnan et al. (2024a). While the absolute value of the cosine similarity is written in Eq. (5) to be 1, it is typically a high value less than 1, where the exact value depends on choices of initialization, architecture, dataset, and training procedure. For more understanding of these conditions, see Beaglehole et al. (2024a).

## D    MODEL AND TRAINING DETAILS

**Gaussian kernel:**    Throughout this work we take bandwidth $L = 2.5$ when using the Mahalanobis Gaussian kernel. We solve ridgeless kernel regression using NumPy on a standard CPU.

**Neural networks:**    Unless otherwise specified, we train one hidden layer neural networks with quadratic activation functions and no biases in PyTorch on a single A100 GPU. Models are trained using AdamW with hidden width 1024, batch size 32, learning rate of $10^{-3}$, weight decay 1.0, and standard PyTorch initialization. All models are trained using the Mean Squared Error loss function (square loss).

For the experiments in Appendix Fig. 5, we train one hidden layer neural networks with quadratic activation and no biases on modular addition modulo $p = 61$. We use 40% training fraction, PyTorch standard initialization, hidden width of $512$, weight decay $10^{-5}$, and AGOP regularizer weight $10^{-3}$. Models are trained with vanilla SGD, batch size 128, and learning rate 1.0.

## E    REORDERING FEATURE MATRICES BY GROUP GENERATORS

Our reordering procedure uses the standard fact of group theory that the multiplicative group $\mathbb{Z}_p^*$ is a cyclic group of order $p - 1$ Koblitz (1994). By definition of the cyclic group, there exists at least one element $g \in \mathbb{Z}_p^*$, known as a *generator*, such that $\mathbb{Z}_p^* = \{g^i \; ; \; i \in \{1, \ldots, p-1\}\}$.

Given a generator $g \in \mathbb{Z}_p^*$, we reorder features according to the map, $\phi_g : \mathbb{Z}_p^* \to \mathbb{Z}_p^*$, where if $h = g^i$, then $\phi_g(h) = i$. In particular, given a matrix $B \in \mathbb{R}^{p \times p}$, we reorder the bottom right $(p-1) \times (p-1)$ sub-block of $B$ as follows: we move the entry in coordinate $(r, c)$ with $r, c \in \mathbb{Z}_p^*$ to coordinate $(\phi_g(r), \phi_g(c))$. For example if $g = 2$ in $\mathbb{Z}_5^*$, then $(2, 3)$ entry of the sub-block would be moved to coordinate $(1, 3)$ since $2^1 = 2$ and $2^3 \bmod 5 = 3$. In the setting of modular multiplication/division, the map $\phi_g$ defined above is known as the *discrete logarithm* base $g$ (Koblitz, 1994, Ch.3). The discrete logarithm is analogous to the logarithm defined for positive real numbers in the sense that it converts modular multiplication/division into modular addition/subtraction. Lastly, in this setting, we note that we only reorder the bottom $(p-1) \times (p-1)$ sub-block of $B$ as the first row and column are 0 (as multiplication by 0 results in 0).

Upon re-ordering the $p \times p$ off-diagonal sub-blocks of the feature matrix by the map $\phi_g$, the feature matrix of RFM for multiplication/division tasks contains circulant blocks as shown in Fig. 3C. Thus, the reordered feature matrices for these tasks also exhibit the structure in Observation 1. As a remark, we note that there can exist several generators for a cyclic group, and thus far, we have not specified the generator $g$ we use for re-ordering. For example, 2 and 3 are both generators of $\mathbb{Z}_5^*$ since $\{2, 2^2, (2^3 \bmod 5), (2^4 \bmod 5)\} = \{3, (3^2 \bmod 5), (3^3 \bmod 5), (3^4 \bmod 5)\} = \mathbb{Z}_5^*$. Lemma K.1 implies that the choice of generator does not matter for observing circulant structure. As a convention, we simply reorder by the smallest generator.

## F    ENFORCING CIRCULANT STRUCTURE IN RFM

We see that the structure in Observation 1 gives generalizing features on modular arithmetic when the circulant $C$ is constructed from the RFM matrix. We observe that enforcing this structure at every iteration, and comparing to the standard RFM model at that iteration, improves test loss and accelerates grokking on e.g. addition (Appendix Fig. 2). The exact procedure to enforce this structure is as follows. We first perform standard RFM to generate feature matrices $M_1, \ldots, M_T$. Then for each iteration of the standard RFM, we construct a new $\widetilde{M}_t$ on which we solve ridgeless kernel

regression for a new $\alpha$ and evaluate on the test set. To construct $\widetilde{M}$, we take $D = \mathbf{diag}\,(M_t)$ and first let $\widetilde{M} = D^{-1/2}MD^{-1/2}$, to ensure the rows and columns have equal scale. We then reset the top left and bottom right sub-matrices of $\widetilde{M}$ as $I - \frac{1}{p}\mathbf{1}\mathbf{1}^T$, and replace the bottom-left and top-right blocks with $C$ and $C^\top$, where $C$ is an exactly circulant matrix constructed from $M_t$. Specifically, where $\boldsymbol{c}$ is the first column of the bottom-left sub-matrix of $M_t$, column $\ell$ of $C$ is equal to $\sigma^\ell(M_t)$.

## G  FOURIER MULTIPLICATION ALGORITHM FROM CIRCULANT FEATURES

As stated in the main text, using certain circulant matrices, kernel regression will learn the Fourier Multiplication Algorithm (FMA). We state the FMA for modular addition/subtraction from Nanda et al. (2023) below. While these prior works write this algorithm in terms of cosines and sines, our presentation simplifies the statement by using the DFT.

**Complex inner product and Discrete Fourier Transform (DFT).**  In our theoretical analysis in Section 5, we will utilize the following notions of complex inner product and DFT. The complex inner product $\langle\cdot,\cdot\rangle_\mathbb{C}$ is a map from $\mathbb{C}^d \times \mathbb{C}^d \to \mathbb{C}$ of the form

$$\langle u, v\rangle_\mathbb{C} = u^\top \bar{v}\,, \tag{6}$$

where $\bar{v}_j$ is the complex conjugate of $v_j$. Let $i = \sqrt{-1}$ and let $\omega = \exp(\frac{-2\pi i}{d})$. The DFT is the map $\mathcal{F} : \mathbb{C}^d \to \mathbb{C}^d$ of the form $\mathcal{F}(u) = Fu$, where $F \in \mathbb{C}^{d\times d}$ is a unitary matrix with $F_{ij} = \frac{1}{\sqrt{d}}\omega^{ij}$. In matrix form, $F$ is given as

$$F = \frac{1}{\sqrt{d}}\begin{pmatrix} 1 & 1 & 1 & \cdots & 1 \\ 1 & \omega & \omega^2 & \cdots & \omega^{d-1} \\ 1 & \omega^2 & \omega^4 & \cdots & \omega^{2(d-1)} \\ \vdots & \vdots & \vdots & \ddots & \vdots \\ 1 & \omega^{d-1} & \omega^{2(d-1)} & \cdots & \omega^{(d-1)(d-1)} \end{pmatrix}. \tag{7}$$

**Fourier Multiplication Algorithm for modular addition/subtraction.**  Consider the modular addition task with $f^*(a, b) = (a + b) \bmod p$. For a given input $x = x_{[1]} \oplus x_{[2]} \in \mathbb{R}^{2p}$, the FMA generates a value for output class $\ell$, $y_{\mathrm{add}}(x; \ell)$, through the following computation:

1. Compute the Discrete Fourier Transform (DFT) for each digit vector $x_{[1]}$ and $x_{[2]}$, which we denote $\widehat{x}_{[1]} = Fx_{[1]}$ and $\widehat{x}_{[2]} = Fx_{[2]}$ where the matrix $F$ is defined in Eq. (7).

2. Compute the element-wise product $\widehat{x}_{[1]} \odot \widehat{x}_{[2]}$.

3. Return $\sqrt{p} \cdot \langle\widehat{x}_{[1]} \odot \widehat{x}_{[2]}, Fe_\ell\rangle_\mathbb{C}$ where $e_\ell$ denotes $\ell$-th standard basis vector and $\langle\cdot,\cdot\rangle_\mathbb{C}$ denotes the complex inner product (see Eq. (6)).

This algorithmic process can be written concisely in the following equation:

$$y_{\mathrm{add}}(x; \ell) = \sqrt{p} \cdot \left\langle Fx_{[1]} \odot Fx_{[2]}, Fe_\ell\right\rangle_\mathbb{C}. \tag{8}$$

Note that for $x = \boldsymbol{e}_a \oplus \boldsymbol{e}_b$, the second step of the FMA reduces to

$$F\boldsymbol{e}_a \odot F\boldsymbol{e}_b = \frac{1}{\sqrt{p}}F\boldsymbol{e}_{(a+b)\bmod p}\,. \tag{9}$$

Using the fact that $F$ is a unitary matrix, the output of the FMA is given by

$$\sqrt{p} \cdot \left\langle \frac{1}{\sqrt{p}}F\boldsymbol{e}_{(a+b)\bmod p}, F\boldsymbol{e}_\ell\right\rangle_\mathbb{C} = \boldsymbol{e}_{(a+b)\bmod p}^\top F^\top \bar{F}\boldsymbol{e}_\ell = \boldsymbol{e}_{(a+b)\bmod p}^\top\boldsymbol{e}_\ell = \mathbb{1}_{\{(a+b)\bmod p=\ell\}}\,. \tag{10}$$

Thus, the output of the FMA is a vector $\boldsymbol{e}_{(a+b)\bmod p}$, which is equivalent to modular addition. We provide an example of this algorithm for $p = 3$ in Appendix J.

**Remarks.** We note that our description of the FMA uses all entries of the DFT, referred to as frequencies, while the algorithm as proposed in prior works allows for utilizing a subset of frequencies. Also note that the FMA for subtraction, written $y_{\text{sub}}$, is similar and given by

$$y_{\text{sub}}(x; \ell) = \sqrt{p} \cdot \left\langle Fx_{[1]} \odot Fe_{p-\ell-1}, Fx_{[2]} \right\rangle_{\mathbb{C}} . \tag{11}$$

Having described the FMA, we now state our theorem.

**Theorem G.1.** *Given all of the discrete data* $\left\{ \left( e_a \oplus e_b, e_{(a-b) \bmod p} \right) \right\}_{a,b=0}^{p-1}$, *for each output class* $\ell \in \{0, \cdots, p-1\}$, *suppose we train a separate kernel predictor* $f_\ell(x) = k(x, X; M_\ell)\alpha^{(\ell)}$ *where* $k(\cdot, \cdot; M_\ell)$ *is a quadratic kernel with* $M_\ell = \begin{pmatrix} 0 & C^\ell \\ (C^\ell)^\top & 0 \end{pmatrix}$ *and* $C \in \mathbb{R}^{p \times p}$ *is a circulant matrix with first row* $e_1$. *When* $\alpha^{(\ell)}$ *is the solution to kernel ridgeless regression for each* $\ell$, *the kernel predictor* $f = [f_0, \ldots, f_{p-1}]$ *is equivalent to Fourier Multiplication Algorithm for modular subtraction (Eq.* (11)).

As $C$ is circulant, $C^\ell$ is also circulant. Hence, each $M_\ell$ has the structure described in Observation 1, where $A = 0$. Note our construction differs from RFM in that we use a different feature matrix $M_\ell$ for each output coordinate, rather than a single feature matrix across all output coordinates. Nevertheless, Theorem G.1 provides support for the fact that block-circulant feature matrices can be used to solve modular arithmetic.

We provide the proof for Theorem G.1 in Appendix K. The argument for the FMA for addition (Eq. (8)) is identical provided we replace $C^\ell$ with $C^\ell R$ and $(C^\ell)^\top$ with $(C^\ell R)^\top$ in each $M_\ell$, where $R$ is the Hankel matrix that reverses the row order (i.e. ones along the main anti-diagonal, zero's elsewhere), whose first row is $e_{p-1}$. An analogous result follows for multiplication and division under re-ordering by a group element, as described in Section 3.

Our proof uses the well-known fact that circulant matrices can be diagonalized using the DFT matrix (Gray et al., 2006) (see Lemma K.2 for a restatement of this fact). This fundamental relation intuitively connects circulant features and the FMA. By using kernels with block-circulant Mahalanobis matrices, we effectively represent the one-hot encoded data in terms of their Fourier transforms. We conjecture that this implicit representation is what enables RFM to learn modular arithmetic with more general circulant matrices when training on just a fraction of the discrete data.

## H GROKKING MULTIPLE TASKS

Throughout the main paper, we focused on modular arithmetic settings for a single task. In more general domains such as language, one may expect there to be many "skills" that need to be learned. In such settings, it is possible that these skills are grokked at different rates. While a full discussion is beyond the scope of this work, to illustrate this behavior, we performed additional experiments in here, where we train RFM on a pair of modular arithmetic tasks simultaneously and demonstrate that different tasks are indeed grokked at different points throughout training.

We train RFM to simultaneously solve the following two modular polynomial tasks: (1) $x + y \bmod p$ ; (2) $x^2 + y^2 \bmod p$ for modulus $p = 61$. We train RFM with the Mahalanobis Gaussian kernel using bandwidth parameter $L = 2.5$. Training data for both tasks is constructed from the same 80% training fraction. In addition to concatenating the one-hot encodings for $x, y$, we also append an extra bit indicating which task to solve (0 indicating task (1) and 1 indicating task (2)). The classification head is shared for both tasks (e.g. output dimension is still $\mathbb{R}^p$).

In Appendix Fig. 3, we observe that there are two sharp transitions in the test loss and test accuracy. By decomposing the loss into the loss per task, we observe that RFM groks task (1) prior to grokking task (2). Overall, these results illustrate that grokking of different tasks can occur at different training iterations.

## I AGOP REGULARIZATION AND WEIGHT DECAY FOR GROKKING MODULAR ARITHMETIC.

It has been argued in prior work that weight decay ($\ell_2$ regularization on network weights) is necessary for grokking to occur when training neural networks for modular arithmetic tasks (Varma et al.,

2023; Davies et al., 2023; Nanda et al., 2023). Under the NFA (Eq. (5)), which states that $W_1^\top W_1$ is proportional to a matrix power of $G(f)$, we expect that performing weight decay on the first layer, i.e., penalizing the loss by $\|W_1\|_F^2 = \text{tr}(W_1^\top W_1)$, should behave similarly to penalizing the trace of the AGOP, $\text{tr}(G(f))$, during training.[3] To this end, we compare the impact of using (1) no regularization; (2) weight decay; and (3) AGOP regularization when training neural networks on modular arithmetic tasks. In Appendix Fig. 5, we find that, akin to weight decay, AGOP regularization leads to grokking in cases where using no regularization results in no grokking and poor generalization. These results provide further evidence that neural networks solve modular arithmetic by using the AGOP to learn features.

## J    FMA EXAMPLE FOR $p = 3$

We now provide an example of the FMA for $p = 3$. Let $x = \boldsymbol{e}_1 \oplus \boldsymbol{e}_2$. In this case, we expect the FMA to output the vector $\boldsymbol{e}_0$ since $(1 + 2) \bmod 3 = 0$. Following the first step of the FMA, we compute

$$\widehat{x}_{[1]} = F\boldsymbol{e}_1 = \frac{1}{\sqrt{3}}[1, \omega, \omega^2]^\top \;\; ; \;\; \widehat{x}_{[2]} = F\boldsymbol{e}_2 = \frac{1}{\sqrt{3}}[1, \omega^2, \omega^4]^\top \;, \tag{12}$$

which are the first and second columns of $F$, respectively. Then their element-wise product is given by

$$F\boldsymbol{e}_1 \odot F\boldsymbol{e}_2 = \frac{1}{3}[1, \omega^3, \omega^6]^\top = \frac{1}{3}[1, 1, 1]^\top = \frac{1}{\sqrt{3}}F\boldsymbol{e}_0 \;, \tag{13}$$

which is $\frac{1}{\sqrt{3}}$ times the first column of the DFT matrix. Finally, we compute the outputs $\sqrt{3} \left\langle \frac{1}{\sqrt{3}}F\boldsymbol{e}_0, F\boldsymbol{e}_\ell \right\rangle_{\mathbb{C}}$ for each $\ell \in \{0, 1, 2\}$. As $F$ is unitary, $y_{\text{add}}(\boldsymbol{e}_1 \oplus \boldsymbol{e}_2; \ell) = \mathbb{1}_{\{1+2=\ell \bmod 3\}}$, so that coordinate 0 of the output will have value 1, and all other coordinates have value 0.

## K    ADDITIONAL RESULTS AND PROOFS

**Lemma K.1.** *Let $C \in \mathbb{R}^{p \times p}$ with its first row and column entries all equal to $0$. Let the $(p - 1) \times (p - 1)$ sub-block starting at the second row and column be $C^\times$. Then, $C^\times$ is either circulant after re-ordering by any generator $q$ of $\mathbb{Z}_p^*$, or $C^\times$ is not circulant under re-ordering by any such generator.*

*Proof of Lemma K.1.* We prove the lemma by showing that for any two generators $q_1, q_2$ of $\mathbb{Z}_p^*$, if $C^\times$ is circulant re-ordering with $q_1$, then it is also circulant when re-ordering by $q_2$.

Suppose $C^\times$ is circulant re-ordering with $q_1$. Let $i, j \in \{1, \ldots, p-1\}$. Note that by the circulant assumption, for all $s \in \mathbb{Z}$,

$$C_{q_1^i, q_1^j} = C_{q_1^{i+s}, q_1^{i+s}} \;, \tag{14}$$

where we take each index modulo $p$.

As $q_2$ is a generator for $\mathbb{Z}_p^*$, we can access all entries of $C^\times$ by indexing with powers of $q_2$. Further, as $q_1$ is a generator, we can write $q_2 = q_1^k$, for some power $k$. Let $a \in \mathbb{Z}$. Then,

$$
\begin{aligned}
C_{q_2^i, q_2^j} &= C_{q_1^{ki}, q_1^{kj}} \\
&= C_{q_1^{ki+ka}, q_1^{kj+ka}} \\
&= C_{q_1^{k(i+a)}, q_1^{k(j+a)}} \\
&= C_{q_2^{i+a}, q_2^{j+a}} \;.
\end{aligned}
$$

Therefore, $C$ is constant on the diagonals under re-ordering by $q_2$, concluding the proof. $\qquad \square$

We next state Lemma K.2, which is used in the proof of Theorem G.1.

---

[3]We note this regularizer been used prior work where AGOP is called the Gram matrix of the input-output Jacobian Hoffman et al. (2019).

**Lemma K.2** (See, e.g., Gray et al. (2006)). *Circulant matrices $U$ can be written (diagonalized) as:*

$$U = F D \bar{F}^\top ,$$

*where $F$ is the DFT matrix, $\bar{F}^\top$ is the element-wise complex conjugate of $F^\top$ (i.e. the Hermitian of $F$), and $D$ is a diagonal matrix with diagonal $\sqrt{p} \cdot F u$, where $u$ is the first row of $U$.*

We now present the proof of Theorem G.1, restating the theorem below for the reader's convenience.

**Theorem.** *Given all of the discrete data $\left\{ \left( e_a \oplus e_b, e_{(a-b) \bmod p} \right) \right\}_{a,b=0}^{p-1}$ in modular subtraction task, for each output class $\ell \in \{0, \cdots, p-1\}$, we train a separate kernel predictor $f_\ell(x) = k(x, X; M_\ell)\alpha^{(\ell)}$. Here $k(\cdot, \cdot; M_\ell)$ is a quadratic kernel with $M_\ell = \begin{pmatrix} 0 & C^\ell \\ (C^\ell)^\top & 0 \end{pmatrix}$ and $C \in \mathbb{R}^{p \times p}$ is a circulant matrix with first row $e_1$. When $\alpha^{(\ell)}$ is the solution to kernel ridgeless regression for each $\ell$, the kernel predictor $f = [f_0, \ldots, f_{p-1}]$ is equivalent to Fourier Multiplication Algorithm for modular subtraction (Eq. (11)).*

*Proof of Theorem G.1.* We present the proof for modular subtraction as the proof for addition follows analogously. We write the standard kernel predictor for class $\ell$ on input $x = x_{[1]} \oplus x_{[2]} \in \mathbb{R}^{2p}$ as,

$$f_\ell(x) = \sum_{a,b=0}^{p-1} \alpha_{a,b}^{(\ell)} k\left( x, e_a \oplus e_b; M_\ell \right) ,$$

where we have re-written the index into kernel coefficients for class $\ell$, $\alpha^{(\ell)} \in \mathbb{R}^{p \times p}$, so that the coefficients are multi-indexed by the first and second digit. Specifically, now $\alpha_{a,b}^{(\ell)}$ is the kernel coefficient corresponding to the representer $k(\cdot, x)$ for input point $x = e_a \oplus e_b$. Recall we use a quadratic kernel, $k(x, z; M_\ell) = (x^\top M_\ell z)^2$. In this case, the kernel predictor simplifies to,

$$f_\ell(x) = \sum_{a,b=0}^{p-1} \alpha_{a,b}^{(\ell)} \left( x_{[1]}^\top C^\ell e_b + e_a^\top C^\ell x_{[2]} \right)^2 .$$

Then, the labels for each pair of input digits, written as a matrix $Y^{(\ell)} \in \mathbb{R}^{p \times p}$ for the $\ell$-th class where the row and column index the first and second digit respectively, are $Y^{(\ell)} = C^{-\ell}$.

For $x = e_{a'} \oplus e_{b'}$, i.e. $x$ in the discrete dataset, we have,

$$
\begin{aligned}
f_\ell(x) &= \sum_{a,b=0}^{p-1} \alpha_{a,b}^{(\ell)} \left( \delta_{(a,b'-\ell)} + \delta_{(a',b-\ell)} + 2\delta_{(a,b'-\ell)} \delta_{(a',b-\ell)} \right) \\
&= e_{b'-\ell}^\top \alpha^{(\ell)} \mathbf{1} + \mathbf{1}^\top \alpha^{(\ell)} e_{a'+\ell} + 2 e_{b'-\ell}^\top \alpha^{(\ell)} e_{a'+\ell} \\
&= e_{b'}^\top C^{-\ell} \alpha^{(\ell)} \mathbf{1} + \mathbf{1}^\top \alpha^{(\ell)} C^{-\ell} e_{a'} + 2 e_{b'}^\top C^{-\ell} \alpha^{(\ell)} C^{-\ell} e_{a'} \\
&= e_{b'}^\top \left( C^{-\ell} \alpha \mathbf{1}\mathbf{1}^\top + \mathbf{1}\mathbf{1}^\top \alpha C^{-\ell} + 2 C^{-\ell} \alpha C^{-\ell} \right) e_{a'} ,
\end{aligned}
$$

where $\delta_{(u,v)} = \mathbb{1}_{\{u=v\}}$. Let $f_\ell(X) \in \mathbb{R}^{p \times p}$ be the matrix of function values of $f_\ell$, where $[f_\ell(X)]_{a,b} = f_\ell(e_a \oplus e_b)$, and, therefore, $f_\ell(e_a \oplus e_b) = e_a^\top f_\ell(X) e_b$. Then, to solve for $\alpha^{(\ell)}$, we need to solve the system of equations for $\alpha$,

$$
\begin{aligned}
f_\ell(X) &= \left( C^{-\ell} \alpha \mathbf{1}\mathbf{1}^\top + \mathbf{1}\mathbf{1}^\top \alpha C^{-\ell} + 2 C^{-\ell} \alpha C^{-\ell} \right)^\top = C^{-\ell} \\
&\iff C^{-\ell} \alpha \mathbf{1}\mathbf{1}^\top + \mathbf{1}\mathbf{1}^\top \alpha C^{-\ell} + 2 C^{-\ell} \alpha C^{-\ell} = C^\ell
\end{aligned}
$$

Note, by left-multiplying both sides by $C^{-\ell}$, we see this equation holds iff,

$$C^{-2\ell} \alpha \mathbf{1}\mathbf{1}^\top + \mathbf{1}\mathbf{1}^\top \alpha C^{-\ell} + 2 C^{-2\ell} \alpha C^{-\ell} = I .$$

Note the solution is unique as the kernel matrix is full rank. We posit the solution $\alpha$ such that $C^{-2\ell} \alpha C^{-\ell} = \frac{1}{2} I + \lambda \mathbf{1}\mathbf{1}^\top$, which is $\alpha = \frac{1}{2} C^{3\ell} + \lambda \mathbf{1}\mathbf{1}^\top$. Then, solving for $\lambda$, we require,

$$\mathbf{1}\mathbf{1}^\top + 2p\lambda \mathbf{1}\mathbf{1}^\top + 2\lambda \mathbf{1}\mathbf{1}^\top = 0 ,$$

which implies $\lambda = -\frac{2}{2p+2}$. Substituting this value of $\lambda$ and simplifying, we see finally that $f_\ell(x) = x_{[1]}^\top C^{-\ell} x_{[2]}$. Therefore, using that circulant matrices are diagonalized by $C = \sqrt{p} F D \bar{F}^\top$ (Lemma K.2) and $\bar{F}^\top F = I$, where $D = \mathbf{diag}(Fe_1)$, we derive,

$$\begin{aligned}
f_\ell(x) &= \sqrt{p} \cdot x_{[1]}^\top F D^{-\ell} \bar{F}^\top x_{[2]} \\
&= \sqrt{p} \cdot x_{[1]}^\top F \mathbf{diag}(Fe_{p-\ell-1}) \bar{F}^\top x_{[2]} \\
&= \sqrt{p} \cdot \left\langle Fx_{[1]} \odot Fe_{p-\ell-1}, Fx_{[2]} \right\rangle_{\mathbb{C}}
\end{aligned}$$

which is the output of the FMA on modular subtraction. $\qquad\square$

## L  LOW RANK SOLUTION TO MODULAR ARITHMETIC

**Addition**  We present a solution to the modular addition task whose AGOP is low rank, in contrast to the full rank AGOP recovered by RFM and neural networks.

We define the "encoding" map $\Phi : \mathbb{R}^p \to \mathbb{C}$ as follows. For a vector $\boldsymbol{a} = [a_0, \ldots, a_{p-1}]$,

$$\Phi(\boldsymbol{a}) = \sum_{k=0}^{p-1} a_k \exp\left(\frac{k 2\pi i}{p}\right) .$$

Notice that $\Phi$ is a linear map such that $\Phi(\boldsymbol{e}_k) = \exp\left(\frac{k 2\pi i}{p}\right)$. Notice also that $\Phi$ is partially invertible with the "decoding" map $\Psi : \mathbb{C} \to \mathbb{R}^p$.

$$\Psi(z) = \widetilde{\max}\left(\left\langle z, \exp\left(\frac{0 \cdot 2\pi i}{p}\right)\right\rangle, \ldots \left\langle z, \exp\left(\frac{(p-1) \cdot 2\pi i}{p}\right)\right\rangle\right) .$$

Above $\widetilde{\max}$ is a function that makes all entries zero except for the largest one and the inner product is the usual inner product in $\mathbb{C}$ considered as $\mathbb{R}^2$. Thus

$$\Psi\left(\exp\left(\frac{k \cdot 2\pi i}{p}\right)\right) = \boldsymbol{e}_k . \tag{15}$$

$\Psi$ is a nonlinear map $\mathbb{C} \to \mathbb{R}^p$. While it is discontinuous but can easily be modified to make it differentiable.

By slight abuse of notation, we will define $\Phi : \mathbb{R}^p \times \mathbb{R}^p \to \mathbb{C}^2$ on pairs:

$$\Phi(\boldsymbol{e}_j, \boldsymbol{e}_k) = (\Phi(\boldsymbol{e}_j), \Phi(\boldsymbol{e}_k)) .$$

This is still a linear map but now to $\mathbb{C}^2$.

Consider now a quadratic map $\boldsymbol{M}$ on $\mathbb{C}^2 \to \mathbb{C}$ given by complex multiplication:

$$\boldsymbol{M}(z_1, z_2) = z_1 z_2 .$$

It is clear that the composition $\Psi \boldsymbol{M} \Phi$ implements modular addition

$$\Psi \boldsymbol{M} \Phi(\boldsymbol{e}_j, \boldsymbol{e}_k) = \boldsymbol{e}_{(j+k) \bmod p}$$

Furthermore, since $\Phi$ is a liner map to a four-dimensional space, the AGOP of the composition $\Psi \boldsymbol{M} \Phi$ is of rank 4.

**Multiplication**  The construction is for multiplication is very similar with modifications which we sketch below. We first re-order the non-zero coordinates by the discrete logarithm with base equal to a generator of the multiplicative group $\boldsymbol{e}_g$ (see Appendix E), while keeping the order of index 0. Then, we modify $\Phi$ to remove index $a_0$ from the sum for inputs $\boldsymbol{a}$. Thus for multiplication,

$$\Phi(\boldsymbol{a}) = \sum_{k=1}^{p-1} a_k \exp\left(\frac{k \cdot 2\pi i}{p-1}\right) ,$$

Hence that $\Phi(\boldsymbol{e}_0) = 0$, $\Phi(\boldsymbol{e}_g) = \exp\left(\frac{2\pi i}{p-1}\right)$ and $\Phi(\boldsymbol{e}_{g^k}) = \exp\left(\frac{k \cdot 2\pi i}{p-1}\right)$. We extend $\Phi$ to $\mathbb{R}^p \times \mathbb{R}^p$ as in Eq. 15 above. Note that $\Phi$ and the re-ordering together are still a linear map of rank 4.

Then, the "decoding" map, $\Psi(z)$, will be modified to return 0, when $z = 0$, and otherwise,

$$\Psi(z) = g^{\widetilde{\max}\left(\left\langle z, \exp\left(\frac{0 \cdot 2\pi i}{p-1}\right)\right\rangle, \ldots \left\langle z, \exp\left(\frac{(p-2) \cdot 2\pi i}{p-1}\right)\right\rangle\right)} .$$

$\boldsymbol{M}$ is still defined as above. It is easy to check that the composition of $\Psi \boldsymbol{M} \Phi$ with reordering implements modular multiplication modulo $p$ and furthermore, the AGOP will also be of rank 4.

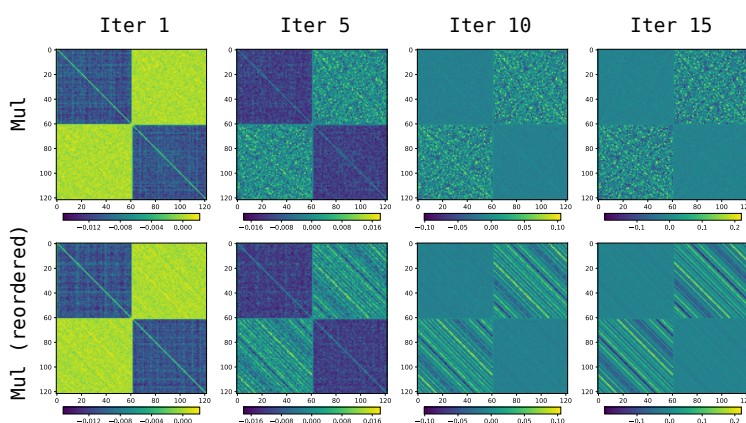

Appendix Figure 1: AGOP evolution for quadratic RFM trained on modular multiplication with $p = 61$ before reordering (top row) and after reordering by the logarithm base 2 (bottom row).

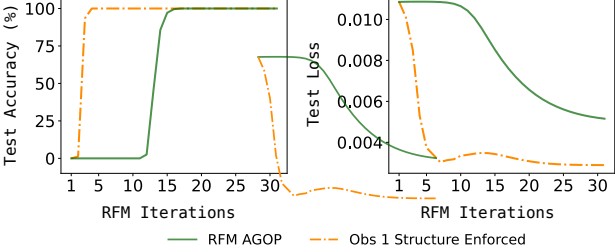

Appendix Figure 2: We train a Gaussian kernel-RFM on $x+y \bmod 97$ and plot test loss and accuracy versus RFM iterations. We also evaluate the performance of the same model upon modifying the $M$ matrix to have exact block-circulant structure stated in Observation 1.

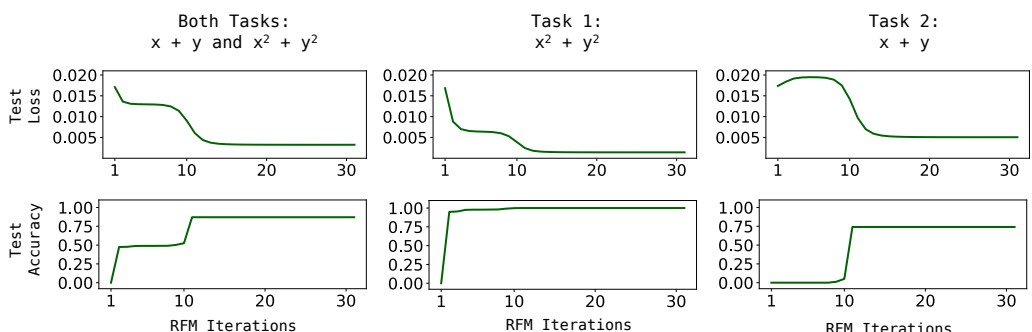

Appendix Figure 3: RFM with the Gaussian kernel trained on two modular arithmetic tasks with modulus $p = 61$. Task 1 is to learn $x^2 + y^2 \bmod p$ and task 2 is to learn $x + y \bmod p$.

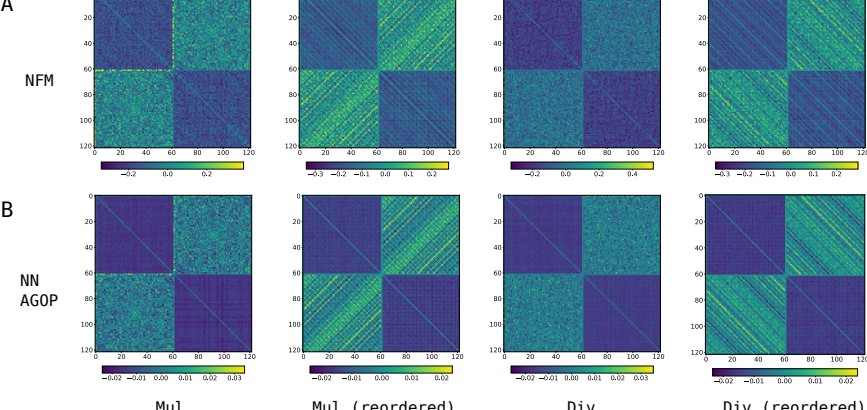

Appendix Figure 4: (A) We visualize the neural feature matrix (NFM) from a one hidden layer neural network with quadratic activations trained on modular multiplication and division, before and after reordering by the discrete logarithm. (B) We visualize the square root of the AGOP of the neural network in (A) before and after reordering.

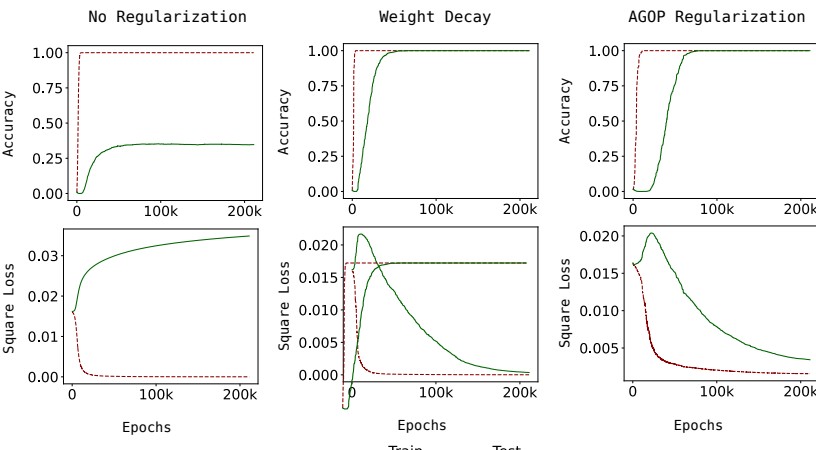

Appendix Figure 5: One hidden layer fully connected networks with quadratic activations trained on modular addition with $p = 61$ with vanilla SGD. Without any regularization the test accuracy does not go to $100\%$ whereas using weight decay or regularizing using the trace of the AGOP result in $100\%$ test accuracy and grokking.

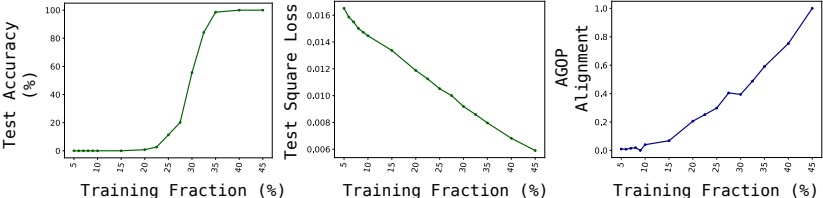

Appendix Figure 6: We train kernel-RFMs for 30 iterations using the Mahalanobis Gaussian kernel for $x + y \bmod 97$. We plot test accuracy, test loss, and AGOP alignment versus percentage of training data used (denoted training fraction). All models reach convergence (i.e., both the test loss and test accuracy no longer change) after 30 iterations. We observe a sharp transition in test accuracy with respect to the training fraction, but we observe gradual change in test loss and AGOP alignment with respect to the training data fraction.

