# OpenReview forum: "Emergence in non-neural models: grokking modular arithmetic via average gradient outer product"
_ICLR.cc/2025/Conference — Submitted to ICLR 2025_

### Official Review · Reviewer_1e5z · 2024-11-03

**Soundness:** 2
**Presentation:** 3
**Contribution:** 2
**Rating:** 5
**Confidence:** 4

**Summary:**

This paper examines the phenomenon of grokking in the context of modular arithmetic tasks. Using Recursive Feature Machines (RFM), which learn features through Average Gradient Outer Product (AGOP) updates rather than gradient descent. The authors show that grokking is not specific to neural networks or gradient descent optimization, but rather arises from feature learning itself. On top of that, the authors introduced different progressive measure that correlates with test accuracies. They also find block-circulant features that implement the Fourier multiplication algorithm.

**Strengths:**

1. Kernel method is rarely discussed in grokking settings
2. The authors discuss their methods clearly
3. Good connection with existing literatures

**Weaknesses:**

1. More discussion on the importance of data is needed:
    - Previous research (e.g., https://arxiv.org/abs/2201.02177) has established that there exists a minimum data threshold required for model grokking. For this kernel-based method, the behavior of this threshold remains unclear. The paper would be significantly strengthened if the authors investigated this threshold using their approach, as their method has fewer hyperparameters, which could provide compelling evidence for the essential role of data quantity in enabling grokking.
    - At least, I would like to see a test-acc vs training data amount curve for the authors' method compared to say GD/Adam training, together with how the learned kernels change with the amount of data during training, especially how the kernels fail to form the correct feature when there is no enough training data.

2. The insight that feature learning is central to grokking is not completely new, for instance, the work in https://arxiv.org/abs/2310.06110 demonstrates the train loss of a neural network decreases much earlier than its test loss can arise due to a neural network
transitioning from lazy training dynamics to a rich, feature learning regime.
3. The mechanism behind the emergence of circular features during algorithm execution remains unclear, which seems crucial for understanding grokking, particularly from a dynamics perspective. From the curves I saw from the paper, I would expect some kind of lazy-to-rich transition happens here as well.

**Questions:**

1. Could the authors provide intuition about how RFM leads to this grokking behavior from the perspective of their iterative algorithm?
2. Is the amount of data required for grokking comparable between this algorithm and other optimization methods like GD/Adam?
3. Does the $p$ play a significant role? What are the implications of using a non-prime modulus?

---

> ### Author Response · Authors · 2024-11-20
>
> We would like to thank the reviewer for their time and insightful comments about our work and address them below. We have included the reviewers' questions in bold for reference.
>
> **_“More discussion on the importance of data is needed. Is the amount of data required for grokking comparable between this algorithm and other optimization methods like GD/Adam?”_**
>
> We have run further experiments comparing test accuracy vs. training data fraction for RFM, kernels with random circulant features, and neural networks trained with AdamW. Figures for these can be seen here: https://imgur.com/a/1TxDY8J.
>
> We will update the final version of our paper with additional discussion about this and include the new plots. We observe that neural networks have somewhat better sample complexity, nevertheless the nature of the transition appears very similar.
>
> **_“The insight that feature learning is central to grokking is not completely new, for instance, the work in https://arxiv.org/abs/2310.06110 demonstrates the train loss of a neural network decreases much earlier than its test loss can arise due to a neural network transitioning from lazy training dynamics to a rich, feature learning regime._**
>
> Regarding the transition from lazy-to-rich dynamics, one of the arguments in our work is that the phenomenon we observe for grokking modular arithmetic in RFM can only be due to feature learning. In our setting the training loss is identically zero and we further observe in the first few iterations that feature learning takes place immediately. For instance, in Figure 2B we observe across addition, subtraction, multiplication, and division that both measures of AGOP alignment and circulant deviation steadily improve at a near-linear rate in the first iterations of RFM in spite of train loss being zero and test loss / accuracy equivalent to random guessing. Thus we do not see evidence for lazy-to-rich transition in RFM training.
>
> **_“The mechanism behind the emergence of circular features during algorithm execution remains unclear, which seems crucial for understanding grokking, particularly from a dynamics perspective. From the curves I saw from the paper, I would expect some kind of lazy-to-rich transition happens here as well._**
>
> **_Could the authors provide intuition about how RFM leads to this grokking behavior from the perspective of their iterative algorithm?”_**
>
> RFM succeeds in feature learning by recursively estimating the target function, then using this estimate to learn a better representation of the data. As the estimated features improve, the estimate of the target function improves, and vice versa. At a high level, we expect that grokking occurs because the error of the learned predictor is a sharp function of the feature quality. While feature quality increases gradually through training (indicated by our progress measures), it is only once the features are estimated to an error below a certain threshold that these features can be utilized to obtain better-than-trivial predictions.
>
> **_“What are the implications of using a non-prime modulus?”_**
>
> Whether or not $p$ is prime does not significantly affect our conclusions. The behaviors we observe for non-prime values are identical for addition and subtraction. For multiplication and division, however, we observe slightly different accuracy curves even though both tasks are able to generalize with RFM. The reason is  that multiplication with non-prime modulus   contains divisors of zero,  e.g., 7, 11, 14…,  for modulus p=77. For division, we must also restrict the training data to those whose denominators are invertible (do not share common divisors with the modulus).
>
> We have run experiments on all four operations for p = 77 using a training fraction of 50% and include a figure of this setting here: https://imgur.com/a/80aPEJB.
>
> We would be happy to include these additional plots in the Appendix of our final paper draft and add an expanded discussion in the main text pointing out this observation.
>
> We once again thank the reviewer for their feedback, if this response addressed your concern please consider raising the score. We look forward to reading the reviewers comments on our rebuttal.

---

> > ### Comment · Reviewer_1e5z · 2024-12-03
> >
> > I appreciate the authors’ new experiments and explanations.
> >
> > My main concern remains as this work does not advance our understanding of the grokking phenomenon. We are at the same stage as the (GD/Adam + modular arithmetic) settings, where the dynamics of feature formation are unclear, and the determinants of sample complexity—such as the significance of the amount of training data presented to the model—are also not well understood.
> >
> > Demonstrating a new setting (even if not a simpler one) that independently exhibits the grokking phenomenon is a meaningful contribution. However, since this work does not advance on the core questions of the grokking phenomenon, I have decided to maintain my score.

---

### Official Review · Reviewer_wQA2 · 2024-11-04

**Soundness:** 3
**Presentation:** 4
**Contribution:** 2
**Rating:** 5
**Confidence:** 4

**Summary:**

The paper attempts to isolate feature-learning in modular arithmetic tasks. They show that the known Fourier-features that are observed in neural networks grokked on modular arithmetic tasks (they call such features Fourier Multiplication Algorithm) can be viewed as "circulant-matrix" structure of input-output Jacobians (called AGOP) along with a global transformation via ridgeless kernel regression. They show that a non-parametric algorithm (RFM) that iteratively refines the kernel using AGOP can grok modular arithmetic tasks and find the aforementioned circulant features. They define progress measures to show the gradual learning of features as well as the agreement between the non-parametric vs neural-network-based training. They also show that initializing network weights using the circulant features accelerates training.

**Strengths:**

- The paper is written in a clear, easy-to-follow manner.
- The supporting experiments are well-presented.
- Studying feature-learning in modular arithmetic tasks with non-parametric methods such as RFM is new. This analysis isolates feature-learning from the neural network training for this task.
- Proofs supporting their claims are presented in the Appendix.

**Weaknesses:**

- Their method is essentially a different way of describing the already-known features in modular arithmetic [1,2]. Continuous progress measures  for this task also already exist in literature, and not new to this work [1,2]. Consequently, the main contribution of this work, in my opinion, is to show that such features can also be reached by using the non-parametric method, viz. RFM. Since the RFM method is also not new to this work [3], I am unable to identify the significant contribution of this paper.
In a barebones way, the message seems to be "The known RFM method applied to modular arithmetic task leads to the known features, as it should."

- The modular addition task $f(a,b) = a + b \\;mod\\; p$ (and other arithmetic tasks after the appropriate re-ordering) respect the modular version of "translation symmetry" in the inputs $a,b$ (i.e. $f(a+t \\;mod\\; p, b-t \\;mod\\; p) = f(a,b)$). Therefore, the feature-matrix of a network that solves this task must also respect this symmetry. This symmetry constraint readily gives the random circular matrix as the feature-matrix, making it trivial in some sense.
(After applying this symmetry, the remaining task is to assign each symmetry class to its correct label, which can be readily achieved by kernel regression.)

**Questions:**

- Do the authors understand why there is a difference in the progress measures between add/sub and mul/div tasks in Figure 5 (B)? Is it related to the fact that mul/sub operations have to deal with 0 separately?

- Can the analysis presented in the work be used to predict the sample complexity of modular arithmetic tasks?

- My understanding is that the circulant features are identical to the Fourier-features found in literature (up to a global transformation learned by kernel regression). Is this assertion correct?

[1] Nanda et al., Progress measures for grokking via mechanistic interpretability (2023)

[2] Gromov, Grokking Modular Arithmetic (2023)

[3] Radhakrishnan et al., Mechanism for feature learning in neural networks and backpropagation-free machine learning models (2024)

---

> ### Author Response · Authors · 2024-11-20
>
> We would like to thank the reviewer for their time and insightful comments about our work and address them below. We have included the reviewers' questions in bold for reference.
>
> **_“Their method is essentially a different way of describing the already-known features in modular arithmetic [1,2]...Since the RFM method is also not new to this work [3], I am unable to identify the significant contribution of this paper...My understanding is that the circulant features are identical to the Fourier-features found in literature (up to a global transformation learned by kernel regression). Is this assertion correct?”_**
>
> We note that the connection to block circulant features is new and has not been observed in prior work. Circulant embeddings are not directly equivalent to Fourier features in neural networks – for example, RFM does not have a notion of a neuron. Further, the feature transformations in neural networks are inherently non-linear, as they consist of a weight matrix composed with an activation function, while circulant matrices perform a strictly linear transformation.
>
> There are many feature maps that can lead to generalization for modular arithmetic and it is, a priori, unclear which feature map set AGOP would recover, if any. We provide an example of another (low rank) solution for modular arithmetic that is recovered by neither RFM nor neural networks in Appendix L.
>
> To the best of our knowledge, there is no other work that has shown a non-parametric model class can learn modular arithmetic tasks. Personally we have found the performance of RFM surprising as a simple kernel machine (the first iteration of RFM) does not perform better than a random guess, yet features improve through RFM iteration.
>
> **_“The feature-matrix of a network that solves this task must also respect this symmetry. This symmetry constraint readily gives the random circular matrix as the feature-matrix, making it trivial in some sense”_**
>
> It is not a priori clear why RFM would learn a symmetry constraint from data, especially given that a simple kernel machine does not perform better than a random guess. Further, even if the generalizing solution found by the model is invariant under this symmetry, it is not clear that the feature matrix learned by RFM satisfies this constraint.
>
> **_“Continuous progress measures for this task also already exist in literature, and not new to this work [1,2]”_**
>
> Thank you for your comment. The progress measures in those works rely on the specific structure of neural networks (neuron specialization) and do not apply to models other than neural networks. In contrast, our measures apply to both neural networks and kernel machines. We will clarify this in the revision.
>
> **_"Do the authors understand why there is a difference in the progress measures between add/sub and mul/div tasks in Figure 5 (B)? Is it related to the fact that mul/sub operations have to deal with 0 separately?”_**
>
> Regarding the difference in progress measures between add/sub and mul/div in Figure 5(B) we do not observe this difference in the trend for RFM. As both RFM and neural networks need to handle the 0 case separately, and RFM does not show this difference, we believe this could be an artifact of the neural network training.
>
> **_“Can the analysis presented in the work be used to predict the sample complexity of modular arithmetic tasks?”_**
>
> Our work provides a relatively simple model, RFM, that exhibits grokking with modular arithmetic. We hope that this model will enable an analysis of sample complexity for these tasks, though it is beyond the scope of our paper. We note that an analysis of sample complexity for neural networks trained by gradient descent has not yet been provided.
>
> We once again thank the reviewer for their feedback, if this response addressed your concern please consider raising the score. We look forward to reading the reviewers comments on our rebuttal.

---

> > ### Comment · Reviewer_wQA2 · 2024-12-03
> >
> > I thank the authors for their detailed responses to my questions and concerns.
> >
> > I maintain my score, which is marginally below acceptance, because I am not convinced that the work provides significant new insights either on modular arithmetic or non-parametric methods such as RFM.
> >
> > However, I will discuss this further with the other reviewers and the area chair with an open mind.

---

### Official Review · Reviewer_7phy · 2024-11-04

**Soundness:** 4
**Presentation:** 4
**Contribution:** 3
**Rating:** 8
**Confidence:** 3

**Summary:**

This paper studies the phenomenon of grokking in non-neural models, specifically Recursive Feature Machines (RFMs). In the setting they study (modular arithmetic), skills emerge purely from feature learning and a block-circulant pattern emerges in the learned square root of the AGOP matrices. Similar to previous work they identify progress measures related to the block circulant structure of the feature matrices which linearly improve during grokking. They also connect their findings to neural networks, showing that the NFM and neural network AGOP exhibit high correlation and contain a similar block circulant structure. Finally, they prove that kernel machines with these block circulant features implement the Fourier multiplication algorithm previously discovered to be implemented by neural networks to solve modular arithmetic.

**Strengths:**

- The paper is written and organized well, with the significance of the results in each section clear.
- While it is a kernel setting, there are several interesting empirical results that cleanly exhibit grokking, the emergence of structured features, and linear progress measures despite training loss being 0 throughout.
- The demonstration of grokking in their kernel setting challenges proposed explanations for grokking, such as grokking having a ‘lazy’ and ‘rich’ regime  with no feature learning and feature learning respectively.
- This work suggests that there is a gap in our available theory to provide effective measures for generalization and explanations for mechanisms of emergence. I believe the work may be valuable to make a call to the community for a need of better theoretical tools and other directions (such as developing progress measures that are more general/applicable without having access to the training mechanism apriori, and are not tied to metrics like loss).

**Weaknesses:**

- This work focuses on a particular task, providing evidence that there may be tasks in which skills can emerge purely as a result of feature learning and independent of loss, but the scope of the results extending to other task and the use case of this particular kernel machine setting is less clear.
- There is still a gap between theory and experiment where there lacks a theoretical proof that grokking will occur, or identify what the mechanism is behind learning modular arithmetic from a finite set of samples.
- I’m not very familiar with the prior literature on RFMs and the use of AGOPs to explain feature learning for neural networks; I know there is context sprinkled throughout in section 2, but it might be useful to reframe Appendix A as a more consolidated review of its introduction and previous use cases.

**Questions:**

1. The empirical validity of emergence has been a popular and contentious point of discussion in the community. Even after Schaeffer’s work, there still does remain tasks which seem emergent independent of metric choice. Do the authors have any thoughts about characteristics of tasks that will predict emergence, or what type of tasks will exhibit emergent behavior?
2. How does the grokking behavior trend with the training fraction?
3. The RFM and NN learn a similar mechanism for the tasks (i.e. implement the Fourier multiplication algorithm)-- in general, to what extent are these learned solutions for tasks exhibiting emergence ‘universal’? Could there be tasks where learned mechanism differs across algorithm or architecture, highlighting a difference in inductive bias? I'm curious if the authors tried training RFMs on other synthetic tasks that have been popular for studying grokking, such as general group operations; I'm wondering if similar structure is observed for eg. group operations on S_5 and S_6, where proposed circuits in neural networks seem to have differing perspectives (learning irreducible representations versus learning subgroup/coset structure [1]).

[1] Stander, Dashiell, et al. "Grokking Group Multiplication with Cosets." arXiv preprint arXiv:2312.06581 (2023).

---

> ### Author Response · Authors · 2024-11-20
>
> We would like to thank the reviewer for their time and insightful comments about our work and address them below. We have included the reviewers' questions in bold for reference.
>
> **_“This work focuses on a particular task, providing evidence that there may be tasks in which skills can emerge purely as a result of feature learning and independent of loss, but the scope of the results extending to other task and the use case of this particular kernel machine setting is less clear.”_**
>
> Like others, we study grokking with modular arithmetic as the task requires feature learning to achieve non-trivial accuracy for non-parametric learning procedures. Further, neural networks and RFM exhibit sharp transitions between trivial and non-trivial error throughout training iterations. In particular, our results are a counterexample to (widely argued) theory that emergence is a “mirage” due to the disconnect between continuous loss function and discrete accuracy. Hence, any explanation for emergence must account for our results, which demonstrate that emergence occurs both in accuracy and loss.
>
> **_“There is still a gap between theory and experiment where there lacks a theoretical proof that grokking will occur, or identify what the mechanism is behind learning modular arithmetic from a finite set of samples.”_**
>
> We note that no prior work establishes a theoretical explanation for how gradient descent induces grokking in real neural networks. Prior works argue that the solutions to modular arithmetic tasks with neural networks satisfy certain margin maximization conditions, while not demonstrating that gradient descent actually performs this maximization. In our work, we provide a proof that kernel machines trained with circulant features can learn the Fourier Multiplication Algorithm (Appendix G), the same generalizing solution found by neural networks.
>
> **_“I’m not very familiar with the prior literature on RFMs and the use of AGOPs to explain feature learning for neural networks; I know there is context sprinkled throughout in section 2, but it might be useful to reframe Appendix A as a more consolidated review of its introduction and previous use cases.”_**
>
> Thank you for the suggestion. We will expand in Appendix A on the introduction of RFM and prior demonstrations of feature learning including deep neural collapse, edge detection in convolutional neural networks, and low-rank matrix completion (to name a few).
>
> **_“How does the grokking behavior trend with the training fraction?”_**
>
> To address the way in which grokking relates to the training fraction we note that Figure 4 and Appendix Figure 6 suggest a scaling law for some values of p by plotting test accuracy vs. training fraction for RFM as well as kernels using random circulant features. To expand on this, we have generated similar plots for additional values of p and a larger set of training fraction percentages to more clearly show the (empirical) relationship between test accuracy and training fraction. In these new figures we have also provided expanded comparisons of RFM to neural networks trained with AdamW and weight decay as well as to kernels with random circulant features. The new figures can be seen here: https://imgur.com/a/1TxDY8J.
>
> We will update the final version of our paper with additional discussion about this and include the new plots. We observe that neural networks have somewhat better sample complexity, nevertheless the nature of the transition appears very similar.
>
> **_“The RFM and NN learn a similar mechanism for the tasks (i.e. implement the Fourier multiplication algorithm)-- in general, to what extent are these learned solutions for tasks exhibiting emergence ‘universal’? Could there be tasks where learned mechanism differs across algorithm or architecture, highlighting a difference in inductive bias?”_**
>
> There is some evidence that the solutions and features for a broader class of arithmetic tasks are similar. For instance he authors of [1] study pretrained LLMs and find that they utilize Fourier features in order to solve arithmetic problems. As we have shown the Fourier Multiplication Algorithm learned by transformers for grokking modular addition is also implemented by the quadratic kernel when trained with 100% of the data (which is given in our theoretical result and we have empirically validated this theorem). This is certainly suggestive of a broader “universal” solution/feature set in the case of arithmetic problems. We also note that these architectures are capable of learning multi-index models (see the discussion in Appendix A) , which is a different and very common setting.  As discussed in the conclusion, unifying these settings remains an open challenge.
>
> We once again thank the reviewer for the feedback.
>
> [1] Tianyi Zhou and Deqing Fu and Vatsal Sharan and Robin Jia, Pre-trained Large Language Models Use Fourier Features to Compute Addition, 2024. arXiv preprint 2406.03445.

---

> ### Author Response · Authors · 2024-11-20
> **Continuation of prior response**
>
> **_“I'm curious if the authors tried training RFMs on other synthetic tasks that have been popular for studying grokking”_**
>
> We have not run experiments on other group structures but we expect that similar grokking will occur. For example it has also been observed that neural networks exhibit hidden progress when learning sparse parities (Barak et al., 2022).

---

> > ### Comment · Reviewer_7phy · 2024-12-03
> >
> > I thank the reviewers for their response and have read the other reviews and responses. Despite the work focusing on one task (modular arithmetic) and one non-parametric method (RFMs), I do believe there is value in the analysis of this setting, where modular arithmetic has grown to be a popular synthetic setting to study grokking, and remain questions which are open even in this domain (eg. sample complexity bounds, or a proof showing that grokking is induced in certain finite-sample settings), and there could be value in having a relatively simple model for future analysis. Thus, I will maintain my score.

---

### Meta-Review · Area_Chair_Sbns · 2024-12-19

**Metareview:**

The authors present an example of grokking on modular arithmetic in recursive feature machines (RFMs), which are neither neural networks nor trained by gradient descent. While grokking has arguably been shown in linear models (https://arxiv.org/abs/2310.16441) it has not been shown with learning methods other than gradient descent. This is interesting and novel, but the reviewers felt that there was not enough novel insight provided in this new setting to justify publication. Part of the beauty of kernel methods is that they are more amenable to interpretable analytic insights that are difficult or impossible with deep learning, and if this paper included theoretical insights about grokking enabled by the new setting, it would go from a borderline to a strong paper. As is, I recommend against acceptance.

**Additional Comments On Reviewer Discussion:**

The reviewers all engaged with the authors during the review period, though none of them changed their mind. Two felt that the paper lacked enough novel insights to justify publishing, while one felt that simply the observation of grokking in a novel setting was enough.

---

### Decision · Program_Chairs · 2025-01-22

Reject